# Self-Aligned Reward: Towards Effective and Efficient Reasoners

**Peixuan Han**[1*]   **Adit Krishnan**[2]   **Gerald Friedland**[2]   **Jiaxuan You**[1]   **Chris Kong**[2†]

[1]University of Illinois Urbana-Champaign    [2]Amazon Web Services
ph16@illinois.edu, luyankon@amazon.com

## Abstract

Reinforcement learning with verifiable rewards has significantly advanced reasoning with large language models (LLMs) in domains such as mathematics and logic. However, verifiable signals provide only coarse-grained or binary correctness feedback. This limitation results in inefficiencies like overly verbose or repetitive reasoning. Existing length-based solutions (e.g., length penalty) compromise accuracy. To address this deficiency, we introduce **self-aligned reward (SAR)**, a generic, universally applicable self-guided signal that complements verifiable rewards to enhance both reasoning accuracy and efficiency in RL. Specifically, SAR is defined as the relative perplexity difference between an answer conditioned on the query and the standalone answer, thereby favoring responses that are concise and query-specific. Quantitative analysis reveals that SAR reliably judges answer quality: concise, correct answers score higher than redundant ones, and partially correct answers score higher than entirely incorrect ones. Evaluation on 4 different models across 7 benchmarks shows that integrating SAR with prevalent RL algorithms like PPO and GRPO reduces answer length by 30%, while improving accuracy by 4%. Our analysis also shows that SAR generalizes well to out-of-domain tasks and achieves a Pareto-optimal frontier between correctness and efficiency compared to state-of-the-art baselines. We also show that SAR shortens unnecessary elaboration while preserving advanced reasoning behaviors. These results highlight the promise of self-aligned reward as a fine-grained complement to verifiable rewards, paving the way for efficient and effective LLM training. All of our code implementations and data are publicly available at Github [1].

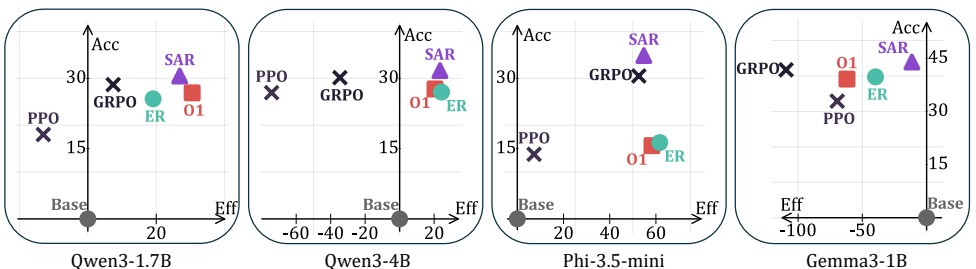

Figure 1: **Training with self-aligned reward enhances both efficiency and accuracy**. We present the relative gains in efficiency and accuracy compared to the respective base model in math reasoning benchmarks. Efficiency gain is measured as the drop in average response length.

## 1 Introduction

Recently, reinforcement learning (RL) with verifiable rewards has attracted broad attention in LLM training, showing remarkable improvements in reasoning skills (Guo et al., 2025; Jaech et al., 2024).

---

[*]Work done during an internship at Amazon Web Services.
[†]Corresponding author.
[1]https://github.com/amazon-science/Self-Aligned-Reward-Towards_
Effective_and_Efficient_Reasoners

Table 1: Comparison of different reward designs.

| Reward | Continuous | Internal | Content-Aware | Correctness | Conciseness |
|---|---|---|---|---|---|
| Correctness | ✗ | ✓ | ✓ | ✓ | ✗ |
| Reward Model (Ouyang et al., 2022) | ✓ | ✗ | ✓ | ✓ | ✗ |
| Length Penalty (Chen et al., 2025a) | ✓ | ✓ | ✗ | ✗ | ✓ |
| Entropy (Agarwal et al., 2025) | ✓ | ✓ | ✓ | ✗ | ✗ |
| **Self-Aligned (Ours)** | ✓ | ✓ | ✓ | ✓ | ✓ |

However, such verifiable signals are inherently discrete and coarse: they only judge final answer correctness, but fail to capture finer distinctions among responses. For instance, an unnecessarily long solution receives no penalty as long as the final answer is correct, and an almost correct response is treated the same as a completely wrong one. This limitation often induces "overthinking", where models generate unnecessary elaborations that increase latency and cost (Sui et al., 2025).

To this end, researchers have proposed heuristic regulations such as length penalties or brevity-oriented objectives (Luo et al., 2025; Aggarwal & Welleck, 2025). While effective in reducing output verbosity, these methods often penalize both redundant and essential reasoning, thereby harming accuracy when necessary intermediate steps are suppressed. Consequently, this line of approaches struggles to balance efficiency with correctness. Using external signal sources, such as reward models, is also undesirable due to their vulnerability to reward hacking. This underscores the necessity of developing internally grounded reward mechanisms that provide precise and detailed guidance, discerning necessary reasoning from redundant elaboration.

To close this gap, we introduce **Self-Aligned Reward (SAR)**, a self-guided proxy to judge answer quality (Equation (6)) based on *perplexity*, an informative metric modeling uncertainty (Friedland et al., 2024; Agarwal et al., 2025). Specifically, SAR evaluates the perplexity of an answer both in isolation and when conditioned on the query, and then measures their relative difference between the two. Consequently, the reward promotes answers that are highly confident under the query context but unlikely to arise independently without the query, which typically corresponds to responses that are concise and strongly aligned with the query. Notably, SAR is the only fine-grained approach that promotes accuracy and efficiency at the same time, as shown in Table 1.

We first conduct a quantitative analysis of different types of answers to demonstrate that SAR provides an accurate fine-grained reward landscape over answers of different qualities (Section 4). We then train LLMs by combining SAR and verifiable reward in PPO and GRPO, two prevalent reinforcement learning algorithms. **We find PPO and GRPO with SAR** (denoted as **SA-PPO** and **SA-GRPO**) **achieve notable gains over baselines across 4 models and 7 benchmarks**, improving accuracy by 4% and efficiency by 30% (Section 5.2). Moreover, SAR outperforms length-based rewards with a Pareto-optimal front in the accuracy-efficiency trade-off (Section 5.3). In addition, we demonstrate the advantages of SAR over confidence-based methods (Section 6.1) and provide an analysis of its reasoning behaviors (Section 6.2). Our findings suggest that combining verifiable rewards with intrinsic model self-judgment offers a new paradigm for RL training, enabling improvements in both reasoning capability and efficiency.

## 2 RELATED WORK

**Reinforcement Learning for LLMs.** Reinforcement learning (RL) has emerged as a powerful paradigm for fine-tuning large language models (LLMs) to enhance their performance in reasoning tasks (Ouyang et al., 2022; Guo et al., 2025). Proximal Policy Optimization (PPO) (Schulman et al., 2017) and Group Relative Policy Optimization (GRPO) (Shao et al., 2024) are typical algorithms that are widely adopted in diverse scenarios and domains (Kulkarni et al., 2024; Chen et al., 2025b; Han et al., 2025a; Liu et al., 2025b). Improvements to these algorithms have also been proposed, namely Dr.GRPO (Liu et al., 2025a), GSPO (Zheng et al., 2025), and Clip-Cov (Cui et al., 2025).

**Efficient Reasoning.** Reasoning models often suffer from overthinking (Su et al., 2025; Cuadron et al., 2025; Sui et al., 2025), leading to unnecessary computation burdens. Prompt engineering (Han et al., 2024; Ma et al., 2025a; Renze & Guven, 2024a) and instruction tuning (Yu et al., 2024; Kang et al., 2025; Xia et al., 2025; Han et al., 2024; Ma et al., 2025b) have been widely used

to mitigate this drawback. Recently, researchers have explored reward signals in RL to penalize lengthy answers (Aggarwal & Welleck, 2025; Team et al., 2025b) or relative lengths (Luo et al., 2025; Arora & Zanette, 2025). In addition, Yeo et al. (2025) explores the relationship of the length penalty with reasoning behaviors and Chen et al. (2025a) proposes a difficulty-sensitive method for token compression. However, these methods sacrifice accuracy to gain efficiency, which is a key issue this paper aims to address.

**Self-judging of LLMs.** The self-judgment capabilities of LLMs (Renze & Guven, 2024b) play a key role in scalable training and self-evolution. For instance, self-confidence estimation is widely used to reduce hallucination (Geng et al., 2023; Wen et al., 2024; Ji et al., 2023), probing internal states provides a lens to analyze model behavior (Han et al., 2025b; Chen et al., 2024), and majority voting (Wang et al., 2022) is critical in inference-time scaling techniques (Snell et al., 2024). Self-judging has also been applied to train reasoning models, with a primary focus on entropy-based metrics (Agarwal et al., 2025; Lei et al., 2025; Zhao et al., 2025). Zhang et al. (2025) provides a theoretical foundation to self-judged signals.

# 3 METHODOLOGY

## 3.1 REINFORCEMENT LEARNING FORMULATION

Proximal Policy Optimization (PPO) (Schulman et al., 2017) and Group Relative Policy Optimization (GRPO) (Shao et al., 2024) are two widely adopted RL algorithms. Given the current policy $\pi_\theta$, query $q$, and the sampled rollout $a$, PPO and GRPO can be formulated as:

$$\mathcal{J}_{\text{PPO}}(\pi_\theta) = \mathbb{E}_{q \sim D, a \sim \pi_\theta(q)} \left[ \frac{1}{|a|} \sum_{j=1}^{|a|} \min \left\{ r_j A_j, \ \text{clip}(r_j, 1-\epsilon, 1+\epsilon) A_j \right\} - \beta \, \text{KL}(\pi_\theta \| \pi_{\text{ref}}) \right], \tag{1}$$

$$\text{where } r_j = \frac{\pi_\theta(a_j \mid q, a_{1\ldots j-1})}{\pi_{\text{old}}(a_j \mid q, a_{1\ldots j-1})}, \ A_j = Q(q, a_{1\ldots j}) - V(q, a_{1\ldots j}). \tag{2}$$

$$\mathcal{J}_{\text{GRPO}}(\pi_\theta) = \mathbb{E}_{q \sim D, \, a_{1\ldots N} \sim \pi_\theta(q)} \left[ \frac{1}{N} \sum_{i=1}^{N} \frac{1}{L_{\max}} \sum_{j=1}^{|a_i|} \min \left\{ r_{i,j} A_i, \ \text{clip}(r_{i,j}, 1-\epsilon, 1+\epsilon) A_i \right\} - \beta \, \text{KL}(\pi_\theta \| \pi_{\text{ref}}) \right], \tag{3}$$

$$\text{where } r_{i,j} = \frac{\pi_\theta(a_{i,j} \mid q, a_{i,1\ldots j-1})}{\pi_{\text{old}}(a_{i,j} \mid q, a_{i,1\ldots j-1})}, \ A_i = R(q, a_i) - \text{mean}[R(q, a_{1\ldots N})]. \tag{4}$$

In the above formulas, $A$ is the advantage, $D$ is the training dataset, and $r$ refers to the importance sampling ratio. $N$, $\epsilon$, $L_{\max}$ and $\beta$ are hyperparameters. Specifically, we use Dr. GRPO (Liu et al., 2025a), an unbiased variant of GRPO in this paper. For verifiable tasks, the reward for PPO and GRPO can be obtained by comparing the model answer with the expected ground truth ($gt$):

$$R_{\text{PPO/GRPO}}(q, a, gt) = R_{\text{VR}}(q, a, gt) = \begin{cases} 1 & \text{if } gt \text{ in } a, \\ 0 & \text{otherwise.} \end{cases} \tag{5}$$

Although RL with verifiable reward has set new benchmarks in reasoning tasks, it induces undesired characteristics such as redundant output due to the coarse binary nature of the reward. To address this issue, two widely used methods introduce targeted penalties on output length: O1-pruner (Luo et al., 2025) and Efficient Reasoner (Arora & Zanette, 2025) (referred to as **O1** and **ER**). Their detailed formulations are provided in Section B.2. Despite their effectiveness, both methods inevitably trade off accuracy, which constrains their broader applicability.

## 3.2 SELF-ALIGNED REWARD

We propose **Self-Aligned Reward (SAR)** (denoted as $R_{\text{SA}}$), an unsupervised holistic signal that combines generation quality, conciseness, and query-answer relevance. The self-aligned reward is estimated entirely by the model policy without any external feedback. We then combine our reward with the binary verifiable reward to address the shortcomings of previous methods, aiming to improve both accuracy and generation efficiency. The reward for self-aligned PPO and self-aligned GRPO (**SA-PPO** and **SA-GRPO**) can be formulated as follows[2]:

$$R_{\text{SA-PPO/GRPO}}(q, a_i, gt) = R_{\text{VR}} + \alpha R_{\text{SA}}, \ R_{\text{SA}} = \text{clip}\left(\frac{\text{ppl}(a_i) - \text{ppl}(a_i|q)}{\text{ppl}(a_i)}, -1, 1\right) \tag{6}$$

$$\text{where ppl}(a) = e^{-\frac{1}{|a|}\sum_{j=1}^{|a|} logP(a_j|a_{1...j-1})}, \ \text{ppl}(a|q) = e^{-\frac{1}{|a|}\sum_{j=1}^{|a|} logP(a_j|q,a_{1...j-1})}. \tag{7}$$

The self-aligned reward captures the relative difference between $\text{ppl}(a)$ and $\text{ppl}(a|q)$, or the **conditioned perplexity drop**, based on the current model policy. Perplexity reflects the likelihood that the model will produce the given response, with lower values indicating higher confidence. Thus, $R_{\text{SA}}$ can be interpreted as: *"How much less likely does the answer become if the query is not present?"*

When an answer is tightly tailored to the query, the conditioned perplexity $\text{ppl}(a|q)$ will be significantly lower than the standalone perplexity $\text{ppl}(a)$, leading to a higher $R_{\text{SA}}$. In contrast, if parts of the answer are irrelevant to the query or include noisy, verbose text, the two perplexities will be similar, leading to a smaller $R_{\text{SA}}$. Therefore, a larger value of $R_{\text{SA}}$ indicates the answer's stronger dependency and better alignment to the query.

Section 4 provides an in-depth analysis of this reward formulation, where we demonstrate that self-aligned reward encourages concise reasoning and effective use of query information, thereby improving both accuracy and training efficiency, and penalizing noisy verbose answers. In addition, the computation of $R_{\text{SA}}$ integrates seamlessly into the RL pipeline with negligible computational overhead, as shown in Sections B and 6.3.

## 4 CASE ANALYSIS: WHY SELF-ALIGNED REWARD WORKS

This section analyzes the self-aligned reward formulation, providing insights on how SAR enhances the reasoning effectiveness and efficiency.

### 4.1 SAR PROVIDES ACCURATE AND FINE-GRAINED SIGNALS

An ideal reward design should be able to rate different types of answers based on their qualities. To assess different reward functions, we analyze 6 different types of answers to 200 math questions sampled from five common math benchmarks (section 5.1) in Table 2. Types **(1)** to **(4)** are obtained by sampling Qwen3-1.7B rollouts at temperature = 1, and using GPT-4o to annotate the responses (see Section C for details). Types **(5)** and **(6)** are artificially synthesized to simulate memorization, where the LLM directly extracts answers from its knowledge without reasoning.

Table 2: We calculate advantage values over the 6 types of answers to the same question ($\alpha = 1$), and report the average over 200 questions. **The advantage values of SA-GRPO accurately rate answers of different qualities.**

| Answer Type | Length | $A_{\text{GRPO}}$ | $A_{\text{GRPO-o1}}$ | $A_{\text{GRPO-ER}}$ | $A_{\text{SA-GRPO}}$ (Ours) |
|---|---|---|---|---|---|
| **(1)**: Correct and concise | 143.5 | 0.5 | 1.04 | 0.81 | 1.15 |
| **(2)**: Correct but redundant | 236.0 | 0.5 | 0.35 | -0.11 | 1.00 |
| **(3)**: Partly correct with wrong answer | 457.1 | -0.5 | -1.28 | -0.65 | -0.01 |
| **(4)**: Completely wrong or irrelevant | 405.8 | -0.5 | -1.19 | -0.65 | -1.44 |
| **(5)**: Correct but no thought | 5.0 | 0.5 | 1.04 | 1.25 | -0.04 |
| **(6)**: Incorrect and no thought | 5.0 | -0.5 | 0.04 | -0.65 | -0.68 |

We present the advantage values (Equation (4)) for each type of response using the following rewards: verifiable GRPO rewards ($A_{\text{GRPO}}$), length-based rewards from the O1 and ER methods ($A_{\text{GRPO-o1}}$, $A_{\text{GRPO-ER}}$) and our self-aligned reward $A_{\text{SA-GRPO}}$. From Table 2, we can observe that SAR provides rich and accurate signals to different answers, exhibiting the following features:

• SAR favors concise and correct answers; it gives a lower reward to long and redundant answers, promoting efficiency. Section 4.2 explains this phenomenon in detail.

• SAR provides partial credit to partly correct answers and penalizes completely irrelevant ones, helping the model learn basic reasoning patterns in the initial stage of training.

---

[2]Since $ppl > 0$, we always have $R_{\text{SA}} < 1$. In practice, $R_{\text{SA}}$ lies in the $(0, 1)$ range in most cases.

• SAR penalizes the synthesized "no thought" answers, even if they're correct and short. This indicates that the reasoning process plays a critical role in SAR, and memorization is discouraged.

On the other hand, the verifiable reward (GRPO) fails to discern answers that are both correct and incorrect, and O1 or ER focus solely on response length, making signals biased and not suitable for maximizing accuracy. For instance, ER isn't applicable to wrong answers, and O1 favors irrelevant answers over partly correct ones, simply because they're shorter. We also provide a qualitative example in Table 12 to illustrate the different answer types.

## 4.2 SAR PROMOTES EXPLOITING QUERY INFORMATION EFFICIENTLY

In this experiment, we take a deeper look at SAR at the token level, aiming to reveal which tokens contribute more to the overall score. Specifically, we decompose the self-aligned reward (Equations (6) and (7)) to calculate the token-level score[3]:

$$R_{\text{SA}} = 1 - \frac{\text{ppl}(a|q)}{\text{ppl}(a)} = 1 - e^{-\frac{1}{|a|}\sum_{j=1}^{|a|} log \frac{P(a_j|q,a_{1\ldots j-1})}{P(a_j|a_{1\ldots j-1})}}.$$

(8)

We then define $v(a_j) = log \frac{P(a_j|q,a_{1\ldots j-1})}{P(a_j|a_{1\ldots j-1})}$ to measure the importance of each token $a_j$ on the whole metric $R_{\text{SA}}$. A token with a higher $v(a_j)$ is considered valuable, while a token with a low or even negative $v(a_j)$ indicates it's less informative and independent of the query.

Figure 2: An illustration of token-level importance scores (i.e. $v(a_j)$). Red color means the token is considered informative for our self-aligned reward, and blue indicates a negative $v(a_j)$. **Tokens extracting new information from the query get high scores.**

From Figure 2, we observe that tokens drawing on information from the question for the first time, like "Janet", "duck", "16", tend to receive high scores. This is because such information is present in the query but absent from previous answer tokens, making $P(a_j|q, a_{1\ldots j-1})$ relatively high while $P(a_j|a_{1\ldots j-1})$ remains low. In contrast, repeating information already generated, like mentioning "Janet" for the second time, results in low scores as both probabilities become similarly high.

Generally, tokens in the earlier part of an answer typically achieve higher $v(a_j)$ values, since extracting new content from the query is easier at the beginning. Later tokens, by comparison, struggle to introduce novel information once much of the query has already been incorporated. This explains why SAR favors short, concise answers and promotes more efficient generation.

## 5 EXPERIMENTS

### 5.1 EXPERIMENTAL SETTINGS

**Datasets.** We utilize five math reasoning benchmarks: GSM8k (Cobbe et al., 2021), MATH (Hendrycks et al., 2021), NuminaMath 1.5 (Li et al., 2024), GSM-symbolic (Mirzadeh et al., 2024), and AIME 1983–2024 (Veeraboina, 2024), covering a difficulty range from primary school to competition level. For training, we combine the training splits of the first three datasets, while the remaining two are not present during training to evaluate generalization.

**Models.** We utilize four base models: Qwen3-1.7B (Base), Qwen3-4B (Base) (Yang et al., 2025), Phi-3.5-mini (Instruct, 3.8B) (Abdin et al., 2024), and Gemma3-1B (Instruct) (Team et al., 2025a), covering different sizes and model families.

---

[3]We omit the clipping for simplicity.

Table 3: Evaluation on math benchmarks. **Self-aligned reward significantly reduces token usage while achieving the best reasoning accuracy**. GRPO-O1 and GRPO-ER results in the table use $\alpha = 0.05$, and SA-GRPO uses $\alpha = 0.2$. Best results among each model are bolded.

| Setting | GSM8k | | GSM-sym | | MATH | | NuminaMath | | AIME | | Average | | |
|---|---|---|---|---|---|---|---|---|---|---|---|---|---|
| | acc | len | acc | len | acc | len | acc | len | acc | len | acc | len | AES |
| **Qwen3-1.7B** | 69.22 | 281 | 46.76 | 365 | 56.89 | 700 | 24.58 | 1242 | 9.97 | 1539 | 41.48 | 825.4 | 0.000 |
| +PPO | 82.71 | 332 | 62.88 | 460 | 63.04 | 784 | 27.02 | 1358 | 9.11 | 1718 | 48.95 | 930.4 | 0.773 |
| **+SA-PPO** | **85.51** | 281 | **68.02** | 369 | 66.27 | 587 | 32.05 | 866 | 10.61 | 1015 | 52.49 | 623.6 | 1.572 |
| +GRPO | 84.53 | 335 | 67.66 | 413 | 67.31 | 697 | 33.44 | 1091 | 13.93 | 1278 | 53.37 | 762.8 | 1.509 |
| +GRPO-O1 | 84.15 | 285 | 65.78 | 363 | 66.51 | 528 | 33.15 | 775 | 13.61 | 910 | 52.64 | 572.2 | 1.652 |
| +GRPO-ER | 82.71 | 244 | 64.74 | 320 | 66.54 | 557 | 34.35 | 980 | 12.22 | 1232 | 52.11 | 666.6 | 1.474 |
| **+SA-GRPO** | **85.51** | 267 | 67.66 | 346 | **67.96** | 564 | **36.03** | 841 | **13.50** | 992 | **54.13** | 602.0 | **1.795** |
| **Qwen3-4B** | 76.19 | 315 | 66.22 | 435 | 63.79 | 652 | 38.80 | 1142 | 20.15 | 1281 | 53.03 | 765.0 | 0.000 |
| +PPO | 91.36 | 373 | 85.42 | 485 | 82.15 | 1132 | 50.80 | 1995 | 26.90 | 2680 | 67.33 | 1333.0 | 0.606 |
| **+SA-PPO** | 92.12 | 266 | 83.52 | 345 | 78.67 | 652 | 49.09 | 1133 | 28.19 | 1397 | 66.32 | 758.6 | 1.260 |
| +GRPO | 92.62 | 320 | 86.30 | 414 | 82.51 | 821 | 53.79 | 1485 | 30.11 | 2113 | 69.07 | 1030.6 | 1.165 |
| +GRPO-O1 | 91.89 | 262 | 86.76 | 350 | 80.41 | 549 | 51.06 | 866 | 28.62 | 1016 | 67.75 | 608.6 | 1.592 |
| +GRPO-ER | 92.80 | 219 | 85.24 | 297 | 79.48 | 503 | 50.47 | 854 | 28.94 | 1022 | 67.38 | 579.0 | **1.596** |
| **+SA-GRPO** | **93.40** | 239 | **87.64** | 323 | **82.63** | 762 | **57.70** | 1358 | **35.69** | 1788 | **71.41** | 894.0 | 1.564 |
| **Phi-3.5-mini** | 71.19 | 287 | 56.86 | 398 | 33.83 | 1132 | 11.56 | 1492 | 2.35 | 943 | 35.15 | 850.4 | 0.000 |
| +PPO | 82.49 | 209 | 64.88 | 322 | 38.56 | 831 | 11.45 | 1707 | **3.85** | 926 | 40.24 | 799.0 | 0.784 |
| **+SA-PPO** | 87.04 | 238 | 73.52 | 344 | 46.34 | 300 | **17.43** | 586 | 2.79 | 501 | 45.42 | 393.8 | 1.997 |
| +GRPO | 87.56 | 235 | **73.76** | 337 | 48.69 | 415 | 16.37 | 582 | 3.21 | 672 | 45.91 | 448.2 | 2.003 |
| +GRPO-O1 | 79.83 | 203 | 62.24 | 279 | 44.41 | 390 | 14.08 | 509 | 2.57 | 583 | 40.62 | 392.8 | 1.316 |
| +GRPO-ER | 82.03 | 154 | 63.54 | 240 | 43.61 | 363 | 12.55 | 511 | 2.47 | 606 | 40.84 | 374.8 | 1.368 |
| **+SA-GRPO** | **87.95** | 207 | 72.72 | 289 | **50.99** | 356 | 16.08 | 455 | 3.22 | 535 | **46.19** | 368.4 | **2.137** |
| **Gemma3-1B** | 42.15 | 325 | 19.58 | 427 | 36.40 | 1042 | 12.40 | 1414 | 2.25 | 2279 | 22.56 | 1097 | 0.000 |
| +PPO | 56.86 | 942 | 30.28 | 1187 | 44.17 | 1408 | 15.57 | 1904 | 1.60 | 2422 | 29.69 | 1572.6 | 1.146 |
| **+SA-PPO** | 55.80 | 683 | 31.00 | 936 | 42.94 | 1025 | 14.22 | 1523 | 1.50 | 1683 | 29.10 | 1170.0 | 1.383 |
| +GRPO | 59.97 | 1208 | **34.70** | 1552 | 45.40 | 1693 | 16.59 | 2264 | 2.25 | 2613 | 31.78 | 1866.0 | 1.343 |
| +GRPO-O1 | 60.80 | 873 | 32.06 | 1157 | 44.76 | 1298 | 16.16 | 1871 | **2.35** | 2449 | 31.23 | 1529.6 | 1.528 |
| +GRPO-ER | 59.44 | 748 | 32.36 | 1063 | 46.23 | 1131 | 16.41 | 1545 | 2.04 | 1696 | 31.29 | 1236.6 | 1.808 |
| **+SA-GRPO** | **61.26** | 552 | 34.52 | 772 | **46.60** | 952 | **16.70** | 1302 | 2.14 | 1509 | **32.24** | 1017.4 | **2.218** |

**Settings.** We train reasoning models with the following methods:

• PPO, GRPO: two traditional RL algorithms, using only the verifiable reward.

• GRPO-O1, GRPO-ER: algorithms with length penalties, aimed to enhance efficiency. See Section B.2 for details.

• SA-PPO, SA-GRPO: algorithms using **self-aligned reward (SAR)**. See Section 3.2 for details.

For training details and hyperparameters, refer to Section B.3.

**Metrics.** In this work, we focus on accuracy and efficiency (measured by average answer length) of LLM reasoning. We report these two metrics as well as an Accuracy-Efficiency trade-off Score (AES). For a trained policy $\pi_\theta$ and its base model $\pi_{\text{ref}}$, we define $\Delta \text{len} = \frac{\text{len}(\pi_{\text{ref}}) - \text{len}(\pi_\theta)}{\text{len}(\pi_{\text{ref}})}$ and $\Delta \text{acc} = \frac{\text{acc}(\pi_\theta) - \text{acc}(\pi_{\text{ref}})}{\text{acc}(\pi_{\text{ref}})}$. Then, $\text{AES}(\pi_\theta) = \Delta \text{len} + \gamma \Delta \text{acc}$ measures the trade-off.[4]

## 5.2 MAIN RESULTS

From Table 3, we observe that **baseline approaches struggle to balance accuracy and efficiency.** PPO and GRPO significantly improve accuracy at the cost of longer and possibly redundant answers. While GRPO-O1 and GRPO-ER effectively reduce computation overhead, they compromise reasoning capability. In contrast, **SAR delivers substantial improvements in both accuracy and efficiency**. Across four base models, SA-GRPO consistently achieves the highest reasoning accuracy while maintaining highly efficient reasoning, with at least 4% improvement in accuracy and 30% reduction in length compared to GRPO. Notably, SA-GRPO produces answers of comparable

---

[4]The choice of $\gamma$ value reflects which aspect does the user prioritize. Since accuracy is the most important factor in most use cases, we set $\gamma = 5$ in practice.

or even shorter length than GRPO-O1 and GRPO-ER, which are explicitly designed for efficient reasoning. A similar phenomenon is observed for SA-PPO, indicating that self-aligned reward applies to diverse RL algorithms. We show two examples comparing GRPO and SA-GRPO in Section F.2.

These findings highlight the superiority of the self-aligned reward mechanism. With a more fine-grained and intelligent signal, SAR makes the model preserve the necessary reasoning which are closely related to query information, and reduces unnecessary content that introduces token burdens, leading to more effective and efficient reasoning.

## 5.3 SELF-ALIGNED REWARD ACHIEVES A PARETO-OPTIMAL IN DYNAMIC BALANCE

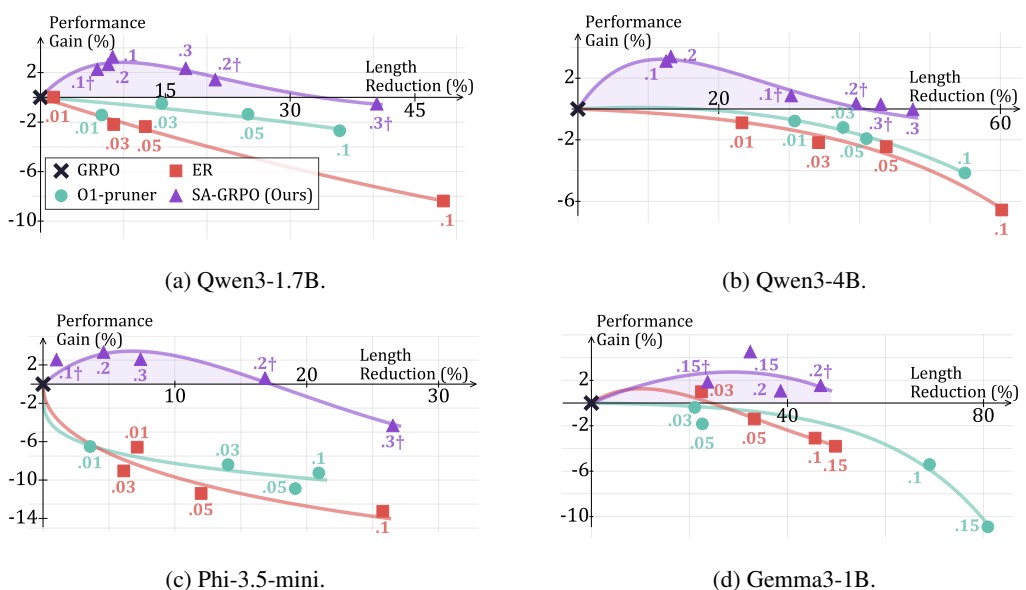

(a) Qwen3-1.7B.  (b) Qwen3-4B.

(c) Phi-3.5-mini.  (d) Gemma3-1B.

Figure 3: Accuracy-efficiency balance of different algorithms. **SA-GRPO reaches a Pareto-optimal curve and achieves notable gains on both axes**. Numbers around each point indicate the $\alpha$ values, and † indicates using 20% of the total training data with the same total training steps.

The hyperparameter $\alpha$ in Equation (6) controls the weights of verifiable reward and self-aligned reward, offering flexibility to focus more on efficiency or accuracy. Therefore, we train SA-GRPO with different $\alpha$ values and compare it with GRPO-O1 and GRPO-ER on 2-dimensional plots.

From Figure 3, we can observe that the curves for SA-GRPO are consistently on the top-right side over curves for GRPO-O1 and GRPO-ER, indicating SA-GRPO balances accuracy and efficiency better than length-based methods. Moreover, length-based methods are always under the x-axis in most cases, showing that these methods sacrifice accuracy for reduced tokens, while SA-GRPO is the only method consistently above the x-axis (illustrated as the light purple area in the figure), showing better accuracy and better efficiency compared to vanilla GRPO. The Pareto-optimal balance between accuracy and efficiency of SA-GRPO further demonstrates the effectiveness of SAR.

## 5.4 SELF-ALIGNED REWARD GENERALIZES TO LOGICAL REASONING

In this section, we examine the generalization ability of self-aligned reward (SAR) to a different domain — logical reasoning - via two benchmark datasets: LogicBench (Parmar et al., 2024) and ProntoQA (Saparov & He, 2022). In Table 4, we find that SA-GRPO improves accuracy compared to GRPO on 5 out of 8 columns, and outperforms length-based methods (GRPO-O1 and ER) on all cases. Similarly, SA-PPO outperforms the PPO baseline on 6 columns. SAR maintains an efficiency gain in the logical domain, not only reducing answer length compared to GRPO and vanilla length-based methods in most cases. These results show that SAR generalizes well to the out-of-domain logical reasoning benchmarks, validating our overall reward design.

Table 4: **SA-PPO and SA-GRPO maintains robustness and efficiency in out-of-domain tasks.**

(a) LogicBench.

| Model | Qwen3 -1.7B | | Qwen3 -4B | | Phi-3.5 -mini | | Gemma3 -1B | |
|---|---|---|---|---|---|---|---|---|
| | acc | len | acc | len | acc | len | acc | len |
| Base | 56.0 | 182 | 75.4 | 336 | 66.0 | 309 | 50.6 | 303 |
| PPO | 60.2 | 340 | 80.0 | 455 | 60.8 | 229 | 48.8 | 264 |
| **SA-PPO** | 64.6 | 368 | 77.6 | 342 | 64.2 | 205 | **55.6** | 241 |
| GRPO | 64.2 | 372 | 78.4 | 416 | **67.6** | 243 | 53.0 | 277 |
| GRPO-O1 | 60.6 | 264 | 76.0 | 350 | 51.0 | 212 | 52.6 | 223 |
| GRPO-ER | 61.4 | 206 | 77.6 | 263 | 49.8 | 143 | 50.0 | 250 |
| **SA-GRPO** | **65.0** | 278 | **80.4** | 208 | 57.4 | 173 | 55.2 | 222 |

(b) ProntoQA.

| Model | Qwen3 -1.7B | | Qwen3 -4B | | Phi-3.5 -mini | | Gemma3 -1B | |
|---|---|---|---|---|---|---|---|---|
| | acc | len | acc | len | acc | len | acc | len |
| Base | 69.0 | 616 | 89.6 | 988 | 94.8 | 339 | 56.2 | 712 |
| PPO | 71.8 | 920 | 97.8 | 561 | 96.4 | 301 | 56.4 | 904 |
| **SA-PPO** | 65.6 | 735 | 99.0 | 405 | **98.2** | 304 | 57.6 | 690 |
| GRPO | 74.0 | 756 | **100.0** | 546 | **98.2** | 254 | 57.4 | 963 |
| GRPO-O1 | 73.0 | 505 | 98.8 | 469 | 87.2 | 227 | 59.0 | 938 |
| GRPO-ER | 73.6 | 424 | 99.0 | 309 | 86.6 | 205 | 58.6 | 919 |
| **SA-GRPO** | **77.8** | 482 | 99.8 | 301 | 90.8 | 216 | **60.6** | 733 |

## 6 ANALYSIS

### 6.1 ABLATION STUDY

In this section, we investigate two critical components of our algorithm that make internal signals effective during training. Firstly, the self-aligned reward (Equation (6)) measures **conditioned perplexity drop**, which is the relative difference between $ppl(a|q)$ and $ppl(a)$, as a proxy for query-answer relevancy. However, existing methods focus mainly on $ppl(a|q)$ alone, which measures entropy or uncertainty(Zhao et al., 2025; Agarwal et al., 2025).Secondly, our approach combines **verifiable reward** with self-aligned reward. We ablate these components and derive the following rewards to compare against SA-GRPO ($R = R_{VR} + \alpha R_{SA}$):

• $R = R_{SA}$: this reward removes the **verifiable reward**, using only the self-aligned reward.

• $R = R_{VR} + \alpha R_{EM} = R_{VR} - \alpha log \, ppl(a|q)$: this reward uses "entropy minimization", a metric measuring self-confidence, as the internal reward, instead of **conditioned perplexity drop**;

• $R = R_{EM} = -log \, ppl(a|q)$: this reward is the entropy minimization reward (Agarwal et al., 2025), ablating both components.

Table 5: GRPO training results on Qwen3-4B with different internal reward signals ($\alpha = 0.2$). **The SA-GRPO formulation obtains optimal performance among baselines.**

| Setting | GSM8k | | GSM-sym | | MATH | | NuminaMath | | AIME | | Average | |
|---|---|---|---|---|---|---|---|---|---|---|---|---|
| | acc | len | acc | len | acc | len | acc | len | acc | len | acc | len |
| Base | 76.19 | 315 | 66.22 | 435 | 63.79 | 652 | 38.80 | 1142 | 20.15 | 1281 | 53.03 | 765.0 |
| $R_{VR}$ | 92.62 | 320 | 86.30 | 414 | 82.51 | 821 | 53.79 | 1485 | 30.11 | 2113 | 69.07 | 1030.6 |
| $R_{EM}$ | 79.15 | 319 | 67.94 | 414 | 68.89 | 970 | 39.86 | 1885 | 21.44 | 2556 | 55.46 | 1228.8 |
| $R_{SA}$ | 39.65 | 87 | 19.54 | 102 | 24.72 | 74 | 16.48 | 80 | 4.39 | 79 | 20.96 | 84.4 |
| $R_{VR} + \alpha R_{EM}$ | 92.25 | 351 | 87.36 | 414 | 81.83 | 779 | 54.78 | 1371 | 33.01 | 1767 | 69.85 | 936.4 |
| $R_{VR} + \alpha R_{SA}$ | **93.40** | 239 | **87.64** | 323 | **82.63** | 762 | **57.70** | 1358 | **35.69** | 1788 | **71.41** | 894.0 |

From Table 5, we can observe that both verifiable signal and conditioned drop measurement are critical for optimal performance. Specifically, $R_{EM}$ shows limited accuracy gain but a large efficiency drop, and $R_{SA}$ converges to shallow reasoning with fewer tokens and poor accuracy . Failure of these methods indicates that the ground-truth signals are still critical for models to develop reasoning skills and ensure training stability, similar to what's found in Zhang et al. (2025)

$R_{VR} + \alpha R_{EM}$ also underperforms SA-GRPO in both accuracy and efficiency, indicating that minimizing entropy is less effective than our approach, which is maximizing conditioned perplexity drop. This is because conditioned perplexity drop provides a more accurate measure of answer quality, as demonstrated in Section 4. Moreover, it avoids issues such as overconfidence or entropy collapse, which can hinder exploration (Zhang et al., 2025; Cui et al., 2025). Overall, the results suggest

"verifiable signal" and "conditioned perplexity drop" are two crucial components in SAR, without which SA-GRPO won't be able to reach optimal performance in accuracy and efficiency.

## 6.2 REASONING BEHAVIORS OF SA-GRPO

Previous work (Gandhi et al., 2025; Zeng et al., 2025) has shown that certain reasoning behaviors are critical to effective and deep reasoning. In this section, we investigate four typical behaviors: backtracking, verification, subgoal setting, and enumeration. Specifically, we use GPT-4o to annotate the reasoning behaviors for models trained with different algorithms (see Section C for details).

Table 6: Frequency of reasoning behaviors. **B**, **V**, **S**, **E** refer to **B**acktracking, **V**erification, **S**ubgoal setting and **E**numeration, respectively. Scores are in percentages. The base model is Qwen3-1.7B. Unlike length-based methods, **SA-GRPO maintains a high usage of reasoning behaviors.**

| Dataset | MATH | | | | NuminaMath | | | | AIME | | | | Average |
|---|---|---|---|---|---|---|---|---|---|---|---|---|---|
| Behavior | **B** | **V** | **S** | **E** | **B** | **V** | **S** | **E** | **B** | **V** | **S** | **E** | |
| Base | 0.8 | 24.2 | 90.8 | 12.2 | 5.4 | 27.0 | 93.4 | 34.8 | 4.4 | 36.2 | 93.2 | 37.4 | 38.0 |
| GRPO | 1.0 | 29.2 | 95.4 | 14.6 | 4.6 | 38.8 | 97.8 | 44.2 | 5.0 | 42.2 | 98.2 | 49.0 | 43.4 |
| GRPO-O1 | 1.2 | 26.6 | 90.4 | 14.6 | 4.8 | 37.0 | 93.2 | 39.6 | 4.2 | 43.4 | 96.0 | 42.8 | 40.6 |
| GRPO-ER | 1.4 | 26.0 | 89.8 | 13.4 | 6.2 | 37.0 | 90.4 | 40.4 | 5.8 | 44.2 | 96.2 | 47.6 | 41.2 |
| **SA-GRPO** | 0.8 | 29.2 | 93.4 | 13.8 | 7.0 | 36.0 | 93.8 | 43.8 | 8.8 | 47.0 | 97.0 | 49.6 | 43.0 |

From Table 6, we can observe that GRPO-O1/ER exhibit fewer reasoning behaviors compared to GRPO, as reasoning behaviors require additional tokens which are penalized by length-based reward functions. However, **SA-GRPO maintains almost the same frequency of reasoning behaviors with GRPO**, notably with 30% fewer tokens than GRPO. This stems from the self-aligned reward's content-aware feature, showcasing its ability to accurately distinguish useful reasoning behaviors from unnecessary content, guiding model behaviors in a more accurate and unbiased manner.

## 6.3 TRAINING COST OF SELF-ALIGNED REWARD

We report the training cost for SA-GRPO in Table 7, from which we can find that SA-GRPO doesn't introduce burdens in "Update" phase compared to vanilla GRPO. The only additional computation in SAR is $ppl(a)$ ($ppl(a|q)$ is already calculated in GRPO, for KL penalty and importance sampling), which only requires a forward pass, making it highly efficient. Additionally, SA-GRPO even

Table 7: Comparison of training time. We report GPU hours (Training time $\times$ GPU count) of training Qwen3-4B for the first 200 steps.

| Method | **Rollout** | **Update** | **Total** |
|---|---|---|---|
| GRPO | 32.95 | 15.13 | 48.08 |
| GRPO-O1 | 30.36 | 15.92 | 46.28 |
| GRPO-ER | 30.92 | 15.24 | 46.16 |
| **SA-GRPO** | 31.44 | 15.20 | 46.64 |

takes less time in the "Rollout" phase due to reduced answer lengths. This suggests that enhancing reasoning efficiency is not only valuable during inference – it can also accelerate RL training and maximize the gain of reasoning capability under limited computation.

## 7 CONCLUSION

In this work, we propose **Self-Aligned Reward (SAR)**, an internal perplexity-based signal evaluating the answer's relevancy with the query, enabling fine-grained supervision beyond binary correctness. Through comprehensive experiments on 4 base models and 7 benchmarks, we demonstrated that SAR enables reinforcement learning to achieve consistent gains of up to 4% in accuracy while reducing response length and computational cost by 30%. Moreover, SAR exhibits a favorable accuracy–efficiency balance compared with length-based baselines, offering a fine-grained and content-aware reward signal that complements verifiable correctness. Our analysis further shows that SAR generalizes robustly to out-of-domain tasks and preserves advanced reasoning behaviors, underscoring its broad applicability. Our findings highlight the significance of incorporating intrinsic fine-grained model self-assessments into the RL framework.

## REPRODUCIBILITY STATEMENT

We ensure the reproducibility of the paper from the following aspects:

• **Dataset**: In Section A, we describe the datasets and their preprocessing methods. All datasets are open-sourced.

• **Method**: Our core contribution is self-aligned reward (SAR), a simple yet effective internal signal. We present the formula and explanations of SAR in Section 3.2, and show the core code for implementing SAR in Section B. In addition, formulations and implementations of the baselines are also presented. Our code is based on VERL, an open-sourced and widely adopted RL framework.

• **Training**: Training configurations are presented in Section B.3. Readers can reproduce the exact training results following these settings.

• **Prompting**: The paper's analytical experiments involve annotating with an LLM. Prompts for such annotation are presented in Section C.

## LLM USAGE STATEMENT

Large language models were not used in the writing of this paper, except for the sample responses from trained LLM models for illustrative purposes. All written content and experimental code were generated solely by the authors.

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

# A DATASETS

We evaluate our approach on five mathematical reasoning datasets with varying difficulty levels, and two logical reasoning datasets. We list the datasets and provide details below.

## A.1 MATHEMATICAL REASONING BENCHMARKS

1. **GSM8k** (Cobbe et al., 2021): A dataset of grade school math word problems.
2. **MATH** (Hendrycks et al., 2021): A challenging dataset of mathematics problems covering various topics.
3. **NuminaMath 1.5** (Li et al., 2024): A comprehensive dataset containing 860k pairs of competition math problems and solutions. We select a subset from the Open-r1 project[5].
4. **GSM-symbolic** (Mirzadeh et al., 2024): A dataset of GSM8k-style problems with different numbers. This dataset is free from data contamination, making it suitable to evaluate generalization ability.
5. **AIME 1983-2024** (Veeraboina, 2024)[6]: A collection of problems from the American Invitational Mathematics Examination spanning over four decades. Requires complex reasoning.

To ensure parsing correctness, we only select questions where the answer is a single integer or fraction number, and remove questions involving geometric plots or hyperlinks. During training, we combined the training splits of GSM8k, MATH, and NuminaMath. GSM-symbolic and AIME were used exclusively for evaluation to test out-of-distribution generalization.

## A.2 LOGICAL REASONING BENCHMARKS

To evaluate the generalization capabilities of our approach beyond mathematical reasoning, we used the following logical reasoning benchmarks:

1. **ProntoQA** (Saparov & He, 2022): A dataset evaluating multi-step syllogistic reasoning.
2. **LogicBench** (Parmar et al., 2024): A comprehensive benchmark for evaluating logical reasoning capabilities of language models across various logical relationships.

These datasets were used only for evaluation purposes and were not part of the training process. Both datasets are in the form of multiple-choice questions, where ProntoQA has 2 choices and LogicBench has 4 for each question.

Table 8 shows the statistics of all datasets used.

## A.3 EVALUATION DETAILS

We construct a unified prompt format for all questions:

> **System Prompt**: You are a reasoning expert assistant. Given a question, you will use your reasoning skills to solve the problem.
> **User Prompt**: **[Question]**
> Please explain your reasoning process before providing an answer.

During evaluation, we parse the last integer or fraction number in the model's output and compare it with the ground truth using `math_verify` package. We don't introduce a strict answer format, as previous work (Zeng et al., 2025) suggest it may hinder exploration.

---

[5] https://huggingface.co/datasets/open-r1/OpenR1-Math-220k
[6] https://www.kaggle.com/datasets/hemishveeraboina/aime-problem-set-1983-2024

Table 8: Sizes of datasets.

| Dataset | GSM8k | MATH | NuminaMath | GSM-sym | AIME | ProntoQA | LogicBench |
|---------|-------|------|-----------|---------|------|----------|-----------|
| # Train | 7473 | 5654 | 10000 | - | - | - | - |
| # Eval | 1319 | 3742 | 2742 | 5000 | 933 | 500 | 500 |

# B  IMPLEMENTATION DETAILS

## B.1  IMPLEMENTATION OF SAR

We implemented all baselines, as well as our approach building on the VERL open-source framework Sheng et al. (2025). The VERL framework provides standard implementations of PPO and GRPO. To implement our approach, we need to calculate the perplexity values for the rollouts given the query ($\mathrm{ppl}(a|q)$) as well as the perplexity of standalone rollouts ($\mathrm{ppl}(a)$).

In most RL algorithms, calculating log probabilities for rollouts are necessary to apply KL penalty. These log-probs can be directly applied to calculate $\mathrm{ppl}(a|q)$ without extra cost:

```python
def compute_ppl(log_probs, mask):
    '''
    log_probs: Tensor[batch_size, seq_len]
    mask: Boolean Tensor[batch_size, seq_len], positions of the answer part are 1; positions
        of query and padding tokens are 0.
    '''
    sum_log_probs = (log_probs * mask).sum(dim=1)
    lengths = mask.sum(dim=1)
    avg_log_probs = sum_log_probs / lengths
    return torch.exp(-avg_log_probs)
```

The calculation of $\mathrm{ppl}(a)$ consists of two steps. The first step is to construct the sequence without the user prompt (we still keep the system prompt):

```python
def construct_empty_question_data(data, tokenizer, q_max_len):
    '''
    data: DataProto, a standard protocol for data exchange in VERL.
    data contains "input_ids", "attention_mask" and "position_ids", which are Tensors[
        batch_size, q_max_len+a_max_len]. Query are left-padded and answers are right-padded,
        which means the answer always begin at index q_max_len.
    '''
    empty_q = f"System:{sys_prompt}\nUser:\nAssistant:"
    empty_q_tokens = tokenizer(empty_q, padding="left", max_length=q_max_len)

    batch_size = data["input_ids"].shape[0]
    for key in ["input_ids", "attention_mask", "position_ids"]:
        data[key][:, :q_max_len] = empty_q_tokens[key].repeat(batch_size, 1)

    last_query_pos = data["position_ids"][:, q_max_len - 1]
    first_answer_pos = data["position_ids"][:, q_max_len] # the first answer token
    gap = first_answer_pos - last_query_pos - 1 # this gap should be zero
    data["position_ids"][:, q_max_len:] -= gap.unsqueeze(-1) # Shift the answer positions
        backward by this gap so they are continuous

    return data
```

The second step is to calculate log probabilities for the "new" sequences. Combining them, we implement the self-aligned reward:

```python
def R_SA(data, tokenizer, q_max_len, policy):
    ppl_qa = compute_ppl(data["log_probs"], data["response_mask"])

    empty_q_data = construct_empty_question_data(data, tokenizer, q_max_len)
    empty_q_log_probs = policy.calc_log_probs(empty_q_data)
    ppl_a = compute_ppl(empty_q_log_probs, data["response_mask"])

    return max((ppl_a - ppl_qa) / ppl_a, -1)
```

In conclusion, the self-aligned reward calculation is fully compatible with VERL's existing optimizations, making it highly efficient for training large models.

## B.2 IMPLEMENTATION OF GRPO-O1 AND GRPO-ER

Signals besides verifiable reward have also been widely studied, where enhancing efficiency is a prevalent direction. In this paper, we consider O1-pruner (Luo et al., 2025) and Efficient Reasoner (ER) (Arora & Zanette, 2025), two typical length-based methods as baselines. Since the original works used offline RL algorithms, we made slight changes to the formula in order to adapt to GRPO:

$$R_{\text{GRPO-o1}}(q, a_i, gt) = R_{\text{VR}} + \alpha R_{\text{o1}}, R_{\text{o1}} = \text{clip}(\frac{\text{mean}[\text{len}(a_{1\dots N})] - \text{len}(a_i)}{\text{len}(a_i)}, -1, 1) \tag{9}$$

$$R_{\text{GRPO-ER}}(q, a_i, gt) = R_{\text{VR}} + \alpha R_{\text{ER}}, R_{\text{ER}} = \begin{cases} 2 * \sigma(\frac{\text{mean}[\text{len}(\mathbf{A}_{\text{cor}})] - \text{len}(a_i)}{\text{std}[\text{len}(\mathbf{A}_{\text{cor}})] + eps}) - 1 & \text{if } R_{\text{VR}} = 1, \\ 0 & \text{otherwise,} \end{cases} \tag{10}$$

where $\sigma(x) = \frac{1}{1+e^{-x}}$ is the Sigmoid function, and $\mathbf{A}_{\text{cor}} = \{a | a \in a_{1\dots N} \wedge R_{\text{VR}}(q, a, gt) = 1\}$ refers to the set of all correct answers.

We also provide the pseudocode for calculating $R_{\text{o1}}$ and $R_{\text{ER}}$ for reference.

```
def R_O1(length: List[int], correctness: List[bool]):
    avg = statistics.mean(length)
    o1_scores = []
    for len in length:
        x = (avg - len) / len
        x = max(min(x, 1), -1)
        o1_scores.append(x)
    return o1_scores

def R_ER(length: List[int], correctness: List[bool]):
    if correctness.count(True) <= 1:
        return [0] * len(length)
    cor_lens = [len for len, correct in zip(length, correctness) if correct]
    avg = statistics.mean(cor_lens)
    std = statistics.stdev(cor_lens)
    er_scores = []
    for len, cor in zip(length, correctness):
        if cor:
            x = (avg - len) / std
            x = 2 / (1 + math.exp(-x)) - 1
            er_scores.append(x)
        else:
            er_scores.append(0)
    return er_scores
```

## B.3 TRAINING SETTINGS

Models are trained on 4 NVIDIA H100 80GB GPUs. For PPO and GRPO, we list all training hyperparameters in Table 9. PPO and SA-PPO use the same config; GRPO, GRPO-O1, GRPO-ER and SA-GRPO also use the same config; the only exception is $\alpha$ in the reward function. The default $\alpha$ for GRPO-O1 and GRPO-ER are 0.05, and 0.2 for SA-GRPO. Using a different $\alpha$ will make these algorithms focus more on accuracy or efficiency, as shown in Figure 3. We use vLLM (Kwon et al., 2023) framework for inference. Models are evaluated using 1 NVIDIA G100 80GB GPU. By default, the maximum response length is 4096, and greedy decoding is used ($\tau = 0$).

Table 9: Training configs for PPO and GRPO.

| Hyperparameter | PPO | GRPO |
|---|---|---|
| Actor learning rate | $1e-6$ | |
| Critic learning rate | $2e-6$ | - |
| train_batch_size[7] | 128 | |
| mini_batch_size | 64 | |
| micro_batch_size | 16 | |
| Training step | 500 | |
| Max response length | 4096 | |
| Num of rollouts | - | 8 |
| Rollout temp ($\tau$) | 1.0 | |
| KL penalty ($\beta$) | $1e-3$ | |
| Advantage clip ($\epsilon$) | 0.2 | |

---

[7]The three batch sizes in the table are hyperparameters in the VERL framework. train_batch_size is the batch size for sampling rollouts, mini_batch_size is the batch size to perform policy updates, and micro_batch_size is the batch size for rollouts and back-propagation. Theoretically, micro_batch_size won't affect the training result, so one can set a smaller or larger value depending on the compute resources.

## C LLM Annotation Prompts

Below is the prompt to categorize responses to 4 types in Section 4:

**System Prompt:** You are an expert reasoner and LLM judge. Given a reasoning problem and an answer, you need to category the answer into one of the following categories:
1: Correct and concise answer. Small mistakes are acceptable.
2: Correct answer, but a bit lengthy, or contains unnecessary steps. Small mistakes are acceptable.
3: Partly correct answers, which makes some mistake and fails to reach the final ground truth.
4: Completely wrong or irrelevant answers, indicating the model doesn't understand the problem.
0: The answer is correct but contains extra content after the answer, like random characters or talking about an irrelevant topic.

**Output Format:**
[thought] Provide your thought process on how you identify the reasoning behaviors.[/thought]
[answer] One single number, indicating the type of the answer. [/answer]

**The one-shot example:**
Query: Making a cake requires 1 cup of flour, 1 cup of sugar, and 2 eggs. Suppose you have 2 cups of flour, 3 cups of sugar, and 6 eggs, how many cakes can you make?
Ground truth answer: 2
Answer: I shall first determine how many cakes each ingredient can support. Flour allows 2 cakes (2/1=2), sugar allows 3 (3/1=3), and eggs allow 3 (6/2=3). Therefore, I can make at most 3 cakes. Wait a second, the maximum number actually depends on the the low-resource ingredient. In this case, it's flour, which only supports 2 cakes. The final answer is 2.

[thought] The answer correctly understands the problem and provides a concise solution. It identifies the limiting ingredient (flour) and calculates the maximum number of cakes that can be made based on that. The reasoning is clear and follows a logical sequence. [/thought]
[answer] 1 [/answer]

Below is the prompt to identify reasoning behaviors in Section 6.2:

**System Prompt:** You are an expert reasoner and LLM judge. Given a reasoning problem and an answer, you need to identify the reasoning behaviors exhibited in the solution. There are four reasoning behaviors that requires identification:
1. **Verification**: This behavior involves systematically checking intermediate results or assumptions to ensure they are correct.
- Example: "Let's verify this result by checking if the two expressions are always equal."
2. **Backtracking**: This behavior occurs when the model explicitly revises its approach after detecting an error or realizing that the current path won't lead to the solution.
- Example: "The assumption that $a > 0$ doesn't work, we need to try something else."
3. **Subgoal Setting**: This behavior involves breaking down a complex problem into smaller, more manageable steps.
- Example: "The first step is to find the range of a and b respectively."
4. **Enumeration**: Solving problems by exhaustively considering multiple cases or possibilities.
- Example: "After investigating 7 days of a week, only Wedensday and Friday satisfies the conditon."

**Guidelines for Identification:**
- It is possible for a single solution to exhibit multiple behaviors or none of them.
- Your annotation should based on the reasoning process, not just the final answer.
- You should only count a behavior if it concretely contributes to the reasoning process.

**Output Format:**
[thought] Provide your thought process on how you identify the reasoning behaviors. [/thought]
[answer] Behavior(s) separated by commas, or "None" if no behavior is identified. [/answer]

# D  ADDITIONAL ANALYSIS AND THEORY

## D.1  GRPO GRADIENT ANALYSIS

To understand how GRPO and our SA-GRPO approach optimize model parameters, we analyze the gradient computation. The gradient of the GRPO objective is:

$$\nabla_\theta \mathcal{J}_{\text{GRPO}}(\theta) = \nabla_\theta \, \mathbb{E}_{q \sim \mathcal{D}, \, \{a_i\}_{i=1}^N \sim \pi_{\theta_{\text{old}}}(\cdot|q)} \left[ \frac{1}{N} \sum_{i=1}^N \frac{1}{|a_i|} \sum_{j=1}^{|a_i|} w_{i,j}(\theta) \, A_i \right] \tag{11}$$

$$= \mathbb{E}_{q \sim \mathcal{D}, \, \{a_i\}_{i=1}^N \sim \pi_{\theta_{\text{old}}}(\cdot|q)} \left[ \frac{1}{N} \sum_{i=1}^N A_i \cdot \frac{1}{|a_i|} \sum_{j=1}^{|a_i|} r_{i,j} \, \nabla_\theta \log \pi_\theta(a_{i,j} \mid q, a_{i,<j}) \right] \tag{12}$$

where

$$r_{i,j} = \frac{\pi_\theta(a_{i,j} \mid q, a_{i,<j})}{\pi_{\theta_{\text{old}}}(a_{i,j} \mid q, a_{i,<j})}, \quad w_{i,j}(\theta) = r_{i,j}.$$

This gradient formulation reveals that GRPO updates model parameters by adjusting the likelihood of generating each token in the rollout responses, weighted by the importance sampling ratio $r_{i,j}$ and the normalized advantage $A_i$. The advantage is calculated based solely on the verifiable reward or other explicit reward functions, without considering the intrinsic quality or relevance of responses.

In standard GRPO, the advantage value $A_i$ is the same for all tokens within a single answer, computed as:

$$A_i = R(q, a_i) - \text{mean}[R(q, a_{1\ldots N})] \tag{13}$$

For correct answers, the advantage is positive, pushing the model to increase the probability of generating such answers. For incorrect answers, the advantage is negative, steering the model away from these outputs.

However, this approach treats all correct answers equally, regardless of their efficiency, relevance, or quality of reasoning. Similarly, it gives all incorrect answers the same negative feedback, missing opportunities to reinforce partially correct reasoning paths or penalize completely irrelevant outputs differently.

## D.2  THEORETICAL ANALYSIS OF SAR

### D.2.1  SELF-ALIGNED REWARD FUNCTION

The key innovation, self-aligned reward $R_{\text{SA}}$, can be derived from the perplexity measures:

$$R_{\text{SA}} = \max \left( \frac{\text{ppl}(a) - \text{ppl}(a|q)}{\text{ppl}(a)}, -1 \right) \tag{14}$$

$$= 1 - \min \left( \frac{\text{ppl}(a|q)}{\text{ppl}(a)}, 2 \right) \tag{15}$$

Expanding with the definitions of perplexity:

$$R_{\text{SA}} = 1 - \min \left( \frac{e^{-\frac{1}{|a|} \sum_{j=1}^{|a|} \log(P(a_j|q, a_{1\ldots j-1}))}}{e^{-\frac{1}{|a|} \sum_{j=1}^{|a|} \log(P(a_j|a_{1\ldots j-1}))}}, 2 \right) \tag{16}$$

$$= 1 - \min \left( e^{-\frac{1}{|a|} \sum_{j=1}^{|a|} [\log(P(a_j|q, a_{1\ldots j-1})) - \log(P(a_j|a_{1\ldots j-1}))]}, 2 \right) \tag{17}$$

$$= 1 - \min \left( e^{-\frac{1}{|a|} \sum_{j=1}^{|a|} \log \left( \frac{P(a_j|q, a_{1\ldots j-1})}{P(a_j|a_{1\ldots j-1})} \right)}, 2 \right) \tag{18}$$

Defining the token-level contribution $v(a_j) = \log\left(\frac{P(a_j|q,a_{1...j-1})}{P(a_j|a_{1...j-1})}\right)$, we get:

$$R_{\text{SA}} = 1 - \min\left(e^{-\frac{1}{|a|}\sum_{j=1}^{|a|} v(a_j)}, 2\right) \tag{19}$$

This formulation reveals that $R_{\text{SA}}$ measures the geometric mean of the likelihood ratio between generating tokens conditioned on the question versus generating them without the question context. Each token $a_j$ contributes $v(a_j)$ to the overall reward.

A positive $v(a_j)$ indicates that token $a_j$ is more likely to be generated when conditioned on the question, suggesting that it leverages information from the query. Conversely, a negative $v(a_j)$ suggests that the token is less likely when conditioned on the question, indicating potential irrelevance or redundancy.

### D.2.2 SA-GRPO Optimization Trajectories

The combined reward $R_{\text{SA-GRPO}}(q, a_i, gt) = R_{\text{VR}} + \alpha R_{\text{SA}}$ leads to a modified advantage calculation:

$$A_i^{\text{SA-GRPO}} = (R_{\text{VR}}(q, a_i, gt) + \alpha R_{\text{SA}}(q, a_i)) - \text{mean}[(R_{\text{VR}}(q, a_{1...N}, gt) + \alpha R_{\text{SA}}(q, a_{1...N}))] \tag{20}$$

During optimization, SA-GRPO's gradient updates follow three key paths:

1. **Correctness Optimization**: Through $R_{\text{VR}}$, SA-GRPO increases the likelihood of generating outputs that contain correct answers, similar to standard GRPO.

2. **Self-Alignment Optimization**: Through $R_{\text{SA}}$, SA-GRPO encourages: - Higher probability for tokens that effectively utilize question information (positive $v(a_j)$) - Lower probability for tokens that are redundant or irrelevant (negative $v(a_j)$) - Concise reasoning by penalizing unnecessary repetition, since repeated information yields low $v(a_j)$ values as it becomes predictable from previous tokens

3. **Memorization Penalty**: SAR naturally penalizes memorized answers. If an answer is memorized, $\text{ppl}(a)$ will be low due to the model's strong prior on the memorized sequence, while the relative drop in perplexity when conditioned on the question would be minimal, resulting in a low or negative $R_{\text{SA}}$.

These optimization paths allow SA-GRPO to simultaneously improve accuracy and efficiency without requiring separate reward models or human preference data. The $\alpha$ hyperparameter controls the balance between correctness and self-alignment objectives, enabling flexible tuning for different accuracy-efficiency trade-offs. As training progresses, we observe from fig. 4:

1. **Early Training Phase**: During early iterations, SA-GRPO primarily optimizes for correctness, as $R_{\text{VR}}$ provides the strongest gradient signal. This establishes a foundation of accurate reasoning.

2. **Mid Training Phase**: Once the model achieves reasonable accuracy, the self-alignment reward $R_{\text{SA}}$ becomes more influential. The optimization begins to focus on improving the efficiency of correct responses by: - Removing tokens with low $v(a_j)$ values (those that don't effectively leverage question information) - Preserving tokens with high $v(a_j)$ values (those that directly address the question)

3. **Late Training Phase**: In the later stages, SA-GRPO fine-tunes the balance between accuracy and efficiency. The combined reward creates a Pareto frontier where further improvements in efficiency come at diminishing costs to accuracy.

Unlike efficiency-focused methods like O1-pruner or Efficient Reasoner, which directly reward shorter responses regardless of content quality, SA-GRPO's optimization is content-aware. It selectively preserves tokens that contribute meaningful information relative to the question, while removing those that don't. In contrast, length-based rewards may inadvertently remove important reasoning steps if they blindly optimize for shorter responses.

The theoretical convergence of SA-GRPO can be expressed as finding the optimal policy $\pi_\theta^*$ that maximizes:

Table 10: SAR results on vision-language models.

| Setting | GSM8k | | GSM-sym | | MATH | | NuminaMath | | AIME | | Average | |
|---|---|---|---|---|---|---|---|---|---|---|---|---|
| | acc | len | acc | len | acc | len | acc | len | acc | len | acc | len |
| **Qwen2-2B-VL** | 14.33 | 1573 | 5.98 | 1873 | 14.94 | 15.84 | 3.65 | 2411 | 0.54 | 2404 | 7.89 | 1969.0 |
| +GRPO | 60.96 | 150 | 27.72 | 209 | 25.25 | 321 | 9.19 | 512 | 0.86 | 561 | 24.80 | 350.6 |
| +GRPO-O1 | 58.38 | 115 | 25.44 | 164 | 23.20 | 79 | 9.99 | 22 | 0.32 | 17 | 23.47 | 79.4 |
| +GRPO-ER | 58.30 | 122 | 26.52 | 179 | 23.73 | 212 | 8.32 | 330 | 0.43 | 383 | 23.46 | 245.2 |
| +SA-GRPO | 59.59 | 121 | 27.66 | 174 | 21.22 | 84 | 9.52 | 56 | 1.29 | 57 | 23.86 | 98.4 |
| **Gemma3-4B** | 74.75 | 264 | 57.40 | 373 | 68.33 | 837 | 35.81 | 1370 | 18.76 | 1843 | 51.01 | 937.4 |
| +GRPO | 89.01 | 409 | 78.68 | 644 | 76.54 | 1177 | 41.9 | 2012 | 18.01 | 2357 | 60.83 | 1355.8 |
| +GRPO-O1 | 87.87 | 292 | 75.02 | 464 | 76.67 | 834 | 41.68 | 1435 | 18.54 | 1814 | 59.95 | 967.8 |
| +GRPO-ER | 88.17 | 338 | 73.36 | 623 | 75.73 | 979 | 41.94 | 1817 | 17.36 | 2268 | 59.31 | 1205.0 |
| +SA-GRPO | 89.16 | 407 | 79.04 | 654 | 75.55 | 1130 | 40.48 | 1920 | 18.11 | 2379 | 60.47 | 1298.0 |

$$\pi_\theta^* = \arg\max_{\pi_\theta} \mathbb{E}_{q \sim \mathcal{D}, a \sim \pi_\theta(\cdot|q)} \left[ R_{\text{VR}}(q, a, gt) + \alpha \max\left( \frac{\text{ppl}(a) - \text{ppl}(a|q)}{\text{ppl}(a)}, -1 \right) \right] \quad (21)$$

At this optimal policy, each generated token in the response contributes maximally to either obtaining the correct answer or efficiently utilizing information from the question, with minimal redundancy or irrelevance.

In practice, this theoretical optimum must balance against the KL divergence penalty that prevents the model from straying too far from the reference policy, ensuring that the learned improvements remain grounded in the model's original capabilities.

# E ADDITIONAL EXPERIMENTS

## E.1 TRAINING TRAJECTORIES

In this section, we show the training trajectories of GRPO, SA-GRPO and SA-GRPO without verifiable reward (a setting discussed in Section 6.1). We report the verifiable reward, self-aligned reward, and average response length at each step.

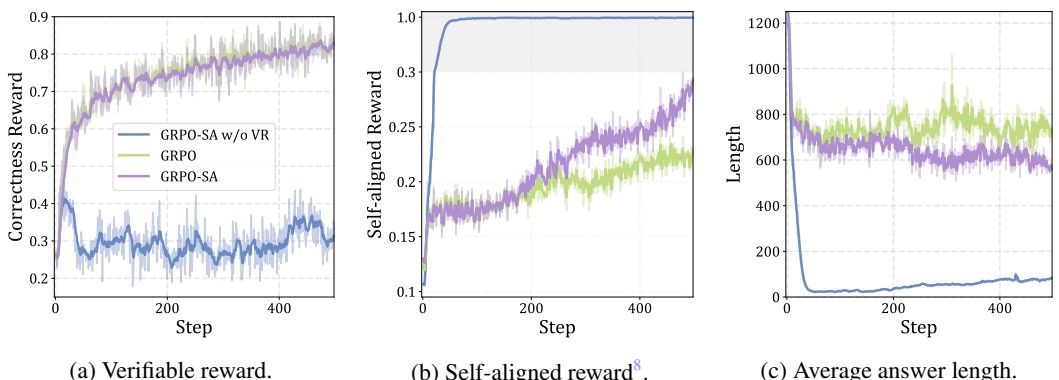

(a) Verifiable reward.  (b) Self-aligned reward[8].  (c) Average answer length.

Figure 4: Training plots for Qwen3-4B.

## E.2 SELF-ALIGNED REWARD ON VISION LANGUAGE MODELS

In this section, we extend SA-GRPO to vision language models.

---

[8]While $R_{\text{SA}}$ isn't used in training the GRPO model, we still calculate and record the values for comparison.

Table 10 summarizes the performance of Qwen2-2B-VL and Gemma3-4B across five reasoning benchmarks under different optimization strategies. Consistent with prior findings, SA-GRPO yields strong gains for purely text-based reasoning—most notably in GSM8k, MATH, and Numina—where its self-aligned reward mechanism effectively tailors outputs to the specific query. For Gemma3-4B, SA-GRPO achieves the highest average accuracy (72.64%), exceeding the best GRPO variant by over one percentage point, while maintaining competitive output lengths. These improvements suggest that SA-GRPO's reward shaping promotes concise, context-aware reasoning paths in textual domains.

However, the results also reveal a notable limitation: SA-GRPO's advantages do not transfer as strongly to vision-language models such as Qwen2-2B-VL. While modest gains are observed in certain benchmarks, its performance is less consistent compared to GRPO, particularly in tasks like GSM-sym and AIME where visual interpretation is coupled with reasoning. We hypothesize that SA-GRPO's self-alignment mechanism, optimized for semantic relevancy in text, is less effective when the query interpretation depends heavily on multimodal fusion. In such cases, reward signals based solely on linguistic alignment may fail to capture errors introduced in the visual grounding stage, leading to weaker overall gains.

These findings suggest that while SA-GRPO is well-suited for text-centric reasoning, its application to multimodal settings may require integrating visual-aware reward components. Future work could explore hybrid reward functions that jointly evaluate semantic correctness and perceptual grounding, enabling SA-GRPO to extend its benefits to vision-language reasoning tasks.

### E.3   EFFECT OF REMOVING THE KL COEFFICIENT

Table 11: Effect of removing the KL penalty. The base model is Qwen3-1.7B in this experiment.

| Setting | GSM8k | | GSM-sym | | MATH | | NuminaMath | | AIME | | Average | |
|---|---|---|---|---|---|---|---|---|---|---|---|---|
| | acc | len | acc | len | acc | len | acc | len | acc | len | acc | len |
| Base | 69.22 | 281 | 46.76 | 365 | 56.89 | 700 | 24.58 | 1242 | 9.97 | 1539 | 41.48 | 825.4 |
| GRPO | 84.53 | 335 | 67.66 | 413 | 67.31 | 697 | 33.44 | 1091 | 13.93 | 1278 | 53.37 | 762.8 |
| GRPO (no KL) | 84.69 | 341 | 69.72 | 430 | 68.31 | 685 | 33.55 | 1121 | 13.61 | 1320 | 53.97 | 779.0 |
| GRPO-O1 | 84.15 | 285 | 65.78 | 363 | 66.51 | 528 | 33.15 | 775 | 13.61 | 910 | 52.64 | 572.2 |
| GRPO-O1 (no KL) | 83.93 | 275 | 66.40 | 356 | 67.13 | 578 | 32.46 | 875 | 14.15 | 1086 | 52.82 | 634 |
| SA-GRPO | 85.51 | 267 | 67.66 | 346 | 67.96 | 564 | 36.03 | 841 | 13.50 | 992 | 54.13 | 602.0 |
| SA-GRPO (no KL) | 85.75 | 246 | 67.62 | 322 | 66.73 | 511 | 34.06 | 788 | 12.43 | 933 | 53.31 | 560 |

In Equation (3), the KL penalty term is used to control the divergence between the reference model and the actor policy, which is critical to preserving the base model's capabilities. We investigate the effect of removing the KL Coefficient (i.e. setting $\beta = 0$ in Equation (3)) in this section.

From Table 11, we observe that for all RL algorithms, the performance gap between using and not using KL is relatively small. This indicates that training remains stable and does not diverge significantly from the base model. It also suggests that the self-aligned signal is sufficiently consistent and well-aligned to guide learning even without strong distributional anchoring.

Notably, removing the KL penalty produces opposite effects for GRPO/GRPO-O1 and SA-GRPO. For GRPO and GRPO-O1, it increases accuracy but reduces efficiency, while for SA-GRPO, it decreases accuracy but improves efficiency. Since the verifiable reward $R_{VR}$ is directly linked to accuracy and the self-aligned reward $R_{SA}$ is not, this pattern suggests that $R_{VR}$ dominates in GRPO-O1 (reward $= R_{VR} + \alpha R_{o1}$), whereas the self-aligned reward $R_{SA}$ dominates in SA-GRPO (reward $= R_{VR} + \alpha R_{SA}$). Across all variants, there's a consistent trade-off between accuracy and efficiency: moving toward higher accuracy tends to reduce efficiency, and vice versa.

# F  CASES

## F.1  DIFFERENT ANSWER TYPES AND ADVANTAGE VALUES

Table 12: An example for different types of answers and their advantage values.

**Query:** Janet's ducks lay 16 eggs per day. She eats three for breakfast every morning and bakes muffins for her friends every day with four. She sells the remainder at the farmers' market daily for $2 per fresh duck egg. How much in dollars does she make every day at the farmers' market?

| Answer Type and Example | Length | $A_{\text{GRPO}}$ | $A_{\text{GRPO-o1}}$ | $A_{\text{GRPO-ER}}$ | $A_{\text{SA-GRPO}}$ (Ours) |
|---|---|---|---|---|---|
| **(1) Correct and Concise:** We can break this down step by step: Janet's ducks lay 16 eggs per day. She eats 3 eggs for breakfast every morning. So, 16 - 3 = 13 eggs remain after Janet eats breakfast. She bakes 4 eggs in muffins every day for her friends. So, we need to subtract these as well: 13 - 4 = 9 eggs. Hence, Janet sells 9 eggs to the farmers' market daily. She sells each egg at $2. Therefore Janet's daily revenue is 9 eggs * $2 = $18. So, Janet makes $18 per day at the farmers' market. | 143 | 0.5 | 0.64 | 0.63 | 0.99 |
| **(2) Correct but redundant:** To answer the question, we need to first calculate how many eggs Janet uses for breakfast and for baking daily. Janet eats 3 eggs for breakfast every morning. Additionally, she bakes muffins with 4 eggs daily. Let's sum the total number of eggs used for both breakfast and baking: 3 (breakfast) + 4 (baking) = 7 eggs total used daily. Now that we know she uses 7 eggs daily, we can determine how many fresh eggs she has left after accounting for her usage. Since her ducks lay 16 eggs per day, we subtract her total egg usage from the daily egg production: 16 (total eggs produced) - 7 (total eggs used) = 9 fresh eggs remaining. Finally, to find out how much money Janet makes selling the fresh eggs, we multiply the number of remaining fresh eggs by the price per egg. She makes $2 per fresh duck egg: 9 (fresh eggs left) × $2 (price per egg) = $18 daily revenue. Therefore, Janet makes $18 per day at the farmers' market, selling the leftover fresh eggs from her ducks. | 236 | 0.5 | 0.29 | -0.29 | 0.84 |
| **(3) Partly Correct, Wrong Answer:** Janet's ducks lay 16 eggs per day. To determine how many fresh eggs she sells each day, we need to subtract the number of eggs she eats for breakfast and the number she uses for muffins: Fresh eggs daily = 16 eggs per day - 3 eggs for breakfast - 4 eggs for muffins = 16 - 3 - 4 = 2 eggs per day So, Janet sells 2 fresh duck eggs daily at the farmers' market. Since she sells each egg for $2, her daily earnings can be calculated by multiplying the number of fresh ducks by their price: Daily earnings = Number of eggs sold x Price per egg = 2 eggs x $2 per egg = $4 Therefore, Janet makes $4 every day at the farmers' market... (Truncated) | 457 | -0.5 | -1.5 | -0.83 | -0.16 |
| **(4) Irrelevant or Completely Wrong:** It is never 30 °C in winter. On the shortest day, in the Arctic Circle, the low is -17 °C. On the longest day, the low is 17 °C. Thus, 10 °C is the average low. We can transform this into a simpler version. Janet's ducks lay 16 eggs per day, eat 3, use 4. Janet makes 13 eggs. She sells them for $0.05 each: 13 × 0.05 = $0.65... (Truncated) | 256 | -0.5 | -0.77 | -0.83 | -0.29 |
| **(5) Correct, but no reasoning (likely memorized):** The answer is 18. | 5 | -0.5 | 1.17 | 1.17 | -0.16 |
| **(6) Incorrect, and no reasoning:** The answer is 0. | 5 | -0.5 | 0.17 | 0.17 | -1.20 |

## F.2 COMPARISON OF GRPO AND SA-GRPO

### F.2.1 CASE 1: IMPROVED CONCISENESS

> **Question**: Let $r$, $s$, and $t$ be solutions to the equation
> $$2x^3 - 3x^2 + 4x - 1 = 0.$$
> Find $r^2 + s^2 + t^2$.
>
> **Ground truth**: $\boxed{-\dfrac{7}{4}}$

**Qwen3-1.7B + GRPO Answer**:

To find $r^2 + s^2 + t^2$ where $r, s$, and $t$ are the solutions to the equation $2x^3 - 3x^2 + 4x - 1 = 0$, we can use the relationships between the roots and the coefficients of the polynomial. Specifically, we can use the identity:

$$r^2 + s^2 + t^2 = (r + s + t)^2 - 2(rs + rt + st)$$

First, we need to find the values of $r + s + t$, $rs + rt + st$, and $rst$ using the coefficients of the polynomial.

The general form of a cubic polynomial is $ax^3 + bx^2 + cx + d = 0$. For our polynomial $2x^3 - 3x^2 + 4x - 1 = 0$, we have:

$$a = 2, \quad b = -3, \quad c = 4, \quad d = -1$$

Using Vieta's formulas, we know:

$$r + s + t = -\frac{b}{a} = -\frac{-3}{2} = \frac{3}{2},$$
$$rs + rt + st = \frac{c}{a} = \frac{4}{2} = 2,$$
$$rst = -\frac{d}{a} = -\frac{-1}{2} = \frac{1}{2}.$$

Now, we can substitute these values into the identity for $r^2 + s^2 + t^2$:

$$r^2 + s^2 + t^2 = (r + s + t)^2 - 2(rs + rt + st)$$

Substituting the values we found:

$$r^2 + s^2 + t^2 = \left(\frac{3}{2}\right)^2 - 2^2$$

Calculating each term:

$$\left(\frac{3}{2}\right)^2 = \frac{9}{4}, \qquad 2^2 = 4$$

So,

$$r^2 + s^2 + t^2 = \frac{9}{4} - 4$$

To subtract these, we need a common denominator:

$$4 = \frac{16}{4}$$

Thus,

$$r^2 + s^2 + t^2 = \frac{9}{4} - \frac{16}{4} = \frac{9 - 16}{4} = \frac{-7}{4}$$

Therefore, the answer is:

$$\boxed{-\frac{7}{4}}$$

### Qwen3-1.7B + SA-GRPO Answer:

To solve this problem, we can use Vieta's formulas, which relate the coefficients of the polynomial to sums and products of its roots. For the polynomial

$$2x^3 - 3x^2 + 4x - 1 = 0,$$

the roots are $r$, $s$, and $t$.

Vieta's formulas tell us:

- The sum of the roots $r + s + t = -\frac{-3}{2} = \frac{3}{2}$.
- The sum of the product of the roots taken two at a time $rs + rt + st = \frac{4}{2} = 2$.
- The product of the roots $rst = -\frac{-1}{2} = \frac{1}{2}$.

We need to find $r^2 + s^2 + t^2$. We can use the identity:

$$r^2 + s^2 + t^2 = (r + s + t)^2 - 2(rs + rt + st).$$

Substituting the values from Vieta's formulas:

$$r^2 + s^2 + t^2 = \left(\frac{3}{2}\right)^2 - 2(2).$$

$$r^2 + s^2 + t^2 = \frac{9}{4} - 4.$$

$$r^2 + s^2 + t^2 = \frac{9}{4} - \frac{16}{4}.$$

$$r^2 + s^2 + t^2 = \frac{9 - 16}{4}.$$

$$r^2 + s^2 + t^2 = \frac{-7}{4}.$$

Therefore, the answer is $\boxed{-\frac{7}{4}}$.

### F.2.2 CASE 2: IMPROVED CONCISENESS AND ACCURACY

> **Question**: How many integers $x$ satisfy
>
> $$|x| + 5 < 7 \quad \text{and} \quad |x - 3| > 2\,?$$
>
> **Ground truth**: $\boxed{2}$

### Qwen3-1.7B + GRPO Answer:

To solve the problem, we need to find the integers $x$ that satisfy both inequalities:

$$1.|x| + 5 < 7 \qquad 2.|x - 3| > 2$$

Let's solve each inequality step by step.

Step 1: Solve the first inequality $|x| + 5 < 7$

Subtract 5 from both sides:
$$|x| < 2$$

This absolute value inequality can be rewritten as:
$$-2 < x < 2$$

So, the integers $x$ that satisfy this inequality are:
$$x = -1, 0, 1$$

Step 2: Solve the second inequality $|x - 3| > 2$

This absolute value inequality can be split into two separate inequalities:
$$x - 3 > 2 \quad \text{or} \quad x - 3 < -2$$

Solving these inequalities:
$$x > 5 \quad \text{or} \quad x < 1$$

So, the integers $x$ that satisfy this inequality are:
$$x = \ldots, -2, -1, 0, 1, 2, 3, 4, 5, \ldots$$

Step 3: Find the intersection of the solutions from both inequalities

We need to find the integers that satisfy both $-2 < x < 2$ and $x > 5$ or $x < 1$.

From the first inequality, the integers are:
$$x = -1, 0, 1$$

From the second inequality, the integers are:
$$x = \ldots, -2, -1, 0, 1, 2, 3, 4, 5, \ldots$$

The intersection of these two sets of integers is:
$$x = -1, 0, 1$$

Step 4: Count the number of integers in the intersection

The integers that satisfy both inequalities are:
$$x = -1, 0, 1$$

There are 3 integers in this set.

Therefore, the number of integers $x$ that satisfy both inequalities is:

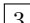

### Qwen3-1.7B + SA-GRPO Answer:

To solve the problem, we need to analyze the two inequalities separately and then find the intersection of the solutions.

Step 1: Solve the first inequality $|x| + 5 < 7$

Subtract 5 from both sides:
$$|x| < 2$$

This means that $x$ must be within 2 units of 0:
$$-2 < x < 2$$

Step 2: Solve the second inequality $|x - 3| > 2$

This inequality can be broken into two separate inequalities:

$$x - 3 > 2 \quad \text{or} \quad x - 3 < -2$$

Solving these, we get:

$$x > 5 \quad \text{or} \quad x < 1$$

Step 3: Find the intersection of the solutions

We need to find the values of $x$ that satisfy both inequalities:

$$-2 < x < 2$$

$$x > 5 \quad \text{or} \quad x < 1$$

The intersection of these intervals is:

$$-2 < x < 1$$

Step 4: Identify the integers in the interval $-2 < x < 1$

The integers in this interval are:

$$-1, 0$$

Step 5: Count the integers

There are 2 integers that satisfy both inequalities.

Therefore, the number of integers $x$ that satisfy the given conditions is:

$$\boxed{2}$$

