# OpenReview forum: "Self-Aligned Reward: Towards Effective and Efficient Reasoners"
_ICLR.cc/2026/Conference — ICLR 2026 Poster_

### Official Review · Reviewer_F8KC · 2025-10-28

**Soundness:** 2
**Presentation:** 3
**Contribution:** 2
**Rating:** 4
**Confidence:** 4

**Summary:**

The paper proposes Self Aligned Reward, SAR, an internal signal that measures the relative perplexity drop between an answer alone and the same answer conditioned on the query. SAR is added to a standard verifiable reward within PPO or GRPO. The goal is to reward answers that are concise and tailored to the query, reducing redundant reasoning while preserving or improving accuracy. Experiments across four small models and seven math and logic benchmarks report about four percent accuracy gains with roughly thirty percent shorter outputs, and a favorable accuracy versus efficiency frontier compared with length penalties such as O1 and ER.

**Strengths:**

- SAR is a small change to existing RL loops and uses only log probabilities already computed, so the added cost is minimal. The paper even reports slightly faster rollouts due to shorter generations.
-  The token level decomposition v(a_j) is a nice lens that explains why early, query specific tokens get rewarded, connecting SAR to information use rather than length alone.
-  Across Qwen3 1.7B and 4B, Phi 3.5 mini, and Gemma3 1B, SAR variants reduce tokens and match or beat accuracy. The Pareto plots make the trade off story visually clear.

**Weaknesses:**

- RSA rises when an answer looks unlikely without the query. This can be increased by echoing or paraphrasing query tokens or by using rare surface forms rather than better reasoning. The qualitative analysis shows discouraging memorized short answers, but an explicit adversarial check would help. Consider adding tests where the model repeats entity strings or injects rare tokens to inflate ppl(a) relative to ppl(a|q).
- GRPO and PPO commonly use a KL term to a reference policy. SAR implicitly prefers answers that are improbable unconditioned on q, which could tug against or amplify KL. The paper should analyze this interaction and show the same KL settings are used across baselines.
- SAR uses the current policy to score both ppl(a) and ppl(a|q). If calibration shifts during training, the reward scale can drift, and the clipping in Equation 6 may hide pathologies. A schedule or normalization across batches might be needed. Please report the distribution of RSA over training and a sensitivity study to the clip range.
- Length penalized GRPO variants often trade accuracy for brevity, yet there are recent methods that make brevity adaptive to difficulty. The paper compares O1 and ER, but please confirm both got per model alpha sweeps as wide as those used for SAR and include best in class brevity methods with per instance stopping rules where applicable. The Pareto plots for SAR vary alpha closely. Ensure comparable sweeps for others

**Questions:**

How sensitive are results to prompt templates that require explicit step by step derivations versus direct short answers? Does SAR over penalize long but necessary derivations when the verifier still rewards only final correctness? Please report token reduction broken down by reasoning behaviors.

Can SAR be computed with teacher forcing on partial prefixes during rollout to provide denser token level rewards, rather than only sequence level returns. This might make the signal less noisy without a learned critic.

What happens if the query contains spurious details? Since SAR rewards dependence on q, models might over attend to red herrings. Add a study with injected distractors.

How do you prevent echoing of the query to raise RSA? Show an explicit penalty or a post hoc filter and quantify its effect.

Do results hold when the base model is much stronger? Most results are for one to four billion parameters. Please include at least a mid size base to test scaling.

---

> ### Author Response · Authors · 2025-11-22
>
> We're very grateful for your constructive comments, and are glad for your acknowledgement of our paper’s clare explanation and strong performance. Below are responses to your suggestions and concerns.
>
> ---
>
> **Q1: What happens if the model repeats entity strings or injects rare tokens to inflate ppl(a) relative to ppl(a|q)?**
>
> We thank the reviewer for raising this thoughtful and important question. This is indeed a potential concern we considered, and we will incorporate this discussion more explicitly in the revised version.
>
> Recall the definition of SAR in **Eq. 6 & 7**. To maximize SAR, the model may either **decrease** $ \mathrm{ppl}(a \mid q) $ or **increase** $ \mathrm{ppl}(a) $. Since we always have $ \mathrm{ppl}(a \mid q) > 1 $, a trivial “hack” to increase SAR would be artificially inflating $ \mathrm{ppl}(a) $ toward infinity. We believe this is precisely why using SAR *alone* performs poorly in **Table 5**.
>
> For the specific failure modes the reviewer mentioned:
>
> - **Injecting rare tokens:** The value $ v(a_j) $ (**Line 223–228**) for an extremely rare token approaches 0, because the token's probability is small both with and without conditioning on $ q $. Thus, adding rare junk tokens *reduces* the average $ v(a_j) $, causing **lower SAR**, not higher.
> - **Repeating strings from the input:** Repeated tokens have **high probability** in both the unconditional and conditional settings, which again drives $ v(a_j) $ toward 0. Repetition, therefore *also lowers* SAR.
>
> Based on this, we emphasize that we do **not** claim SAR to be a standalone unsupervised method. Instead, **SAR is designed as a complement to rule-based rewards**, improving reasoning efficiency and clarity without enabling trivial reward hacking.
>
> ---
>
> **Q2:  What’s the interaction between SAR and KL penalty? The author should show that the same KL settings are used across baselines.**
>
> We believe SAR and the KL penalty operate in largely orthogonal dimensions and do not directly interfere, since **SAR acts horizontally, while KL acts vertically.** Let the policy at step $ k $ be $ \pi_k $. SAR compares **within-step distributions:** $ \pi_k(a) $ vs. $ \pi_k(a \mid q) $, while  KL compares **across-step distributions:** $ \pi_k(a \mid q) $ vs. $ \pi_{k-1}(a \mid q) $ (or vs. $ \pi_0(a \mid q) $). Thus, SAR shapes the content-quality dimension, while KL regulates policy drift, and their effects are independent.
>
> In addition, as shown in **App. E.3 (Table 11)**, SA-GRPO performs very stably both with and without KL penalties. This empirical evidence supports that the two mechanisms do not negatively interact.
>
> ---
>
> **Q3: Could the update of the policy affect the clipping of SAR? Please report the distribution of SAR over training and a sensitivity study to the clip range.**
>
> We would like to emphasize that the clipping in **Eq. 6** is primarily for mathematical completeness. In actual rollouts, clipping is **extremely rare**, and SAR values almost never become negative.
>
> SAR becomes negative only when $ \mathrm{ppl}(a) < \mathrm{ppl}(a \mid q) $, but since answers are always sampled from $ \pi(a \mid q) $, and as long as the base model is reasonably grounded, we almost never observe such cases in practice.
>
> Furthermore, the reward trajectories shown in **Figure 4** support this claim. In particular:
>
> - The **purple line** (GRPO-SA) shows that both correctness reward and SAR increase **smoothly and steadily** during training.
> - There is no observable instability or plateauing that would indicate clipping effects or pathological SAR behavior.

---

> ### Author Response · Authors · 2025-11-22
>
> **Q4: Does SAR sacrifice accuracy for brevity? Does the author consider methods that make brevity adaptive to difficulty? How is the alpha value range chosen in the curve?**
>
> 1. **SAR does not trade accuracy for brevity.**
>
>     Traditional length-penalized objectives often reduce answer length by directly punishing tokens, which can force the model to discard necessary reasoning and thereby hurt accuracy. By contrast, **SAR is content-aware**: it incorporates perplexity into the reward so that brevity is rewarded *only when the concise output remains informative and strongly conditioned on the question*. This design prevents the model from preferring shorter but incorrect answers; instead, SAR encourages **precise and substantively meaningful brevity**.
>
> 2. **Adaptive brevity methods are complementary to SAR.**
>
>     We are aware of recent approaches that adapt output length based on task difficulty (e.g., per-instance stopping rules) **[1]**. Those methods operate at inference time to control output length, while SAR is a *training-time*, reward-design approach that aligns the optimization objective toward efficient reasoning. Because they act on different mechanisms, these techniques are **complementary**: in principle, difficulty-adaptive inference could be combined with SAR to yield further gains in efficiency, but they are not directly comparable as training-time baselines and therefore were not included in the main comparisons.
>
> 3. **How α ranges are chosen for Pareto curves.**
>
>     Reward functions differ in scale and sensitivity, so using identical α numerical ranges across methods can be unfair. For each method we therefore perform an **α sweep within the widest interval that preserves stable training**. Concretely, we increase/decrease α until we observe unstable gradient norms or performance collapse, then stop extending the sweep in that direction. Each curve in **Figure 3** therefore shows the best attainable Pareto frontier for that method under its **maximally stable α range**. This procedure yields a fair comparison of each method’s achievable accuracy/efficiency trade-offs.
>
>
> ---
>
> **Q5: How sensitive are results to prompts? Does SAR over penalize long but necessary derivations when the verifier still rewards only final correctness? Can token reduction broken down by reasoning behaviors be calculated?**
>
> 1. After RL training, the model’s policy is mostly fixed and isn’t sensitive to prompts. For the base model, we conduct an experiment that adds the following sentence to encourage brevity:
>
> ```
> Please keep your outputs simple and concise, and get the answer without any pause, check or reflection.
> ```
>
> The table below reports (accuracy – average token length) for the baseline prompt vs. the briefness prompt. Differences are small, indicating that prompt-level nudges have limited effect compared to training-time rewards.
>
> | Base | Prompt | GSM | GSM-sym | MATH | NuminaMath | AIME |
> | --- | --- | --- | --- | --- | --- | --- |
> | Qwen3-1.7B | Normal | 69.22 - 281 | 46.76 - 365 | 56.89 - 700 | 24.58 - 1242  | 9.97 - 1539 |
> | Qwen3-1.7B | Brief | 68.84 - 288 | 46.28 - 365 | 57.00 - 712 | 26.44 - 1251 | 8.79 - 1527 |
> | Qwen3-4B | Normal | 76.19 - 315 | 66.22 - 435 | 63.79 - 652 | 38.80 - 1142 | 20.15 - 1281 |
> | Qwen3-4B | Brief | 75.82 - 333 | 66.56 - 407 | 62.67 - 643 | 38.73 - 1149 | 19.82 - 1284 |
> | Phi-3.5-mini | Normal | 71.19 - 287 | 56.86 - 398 | 33.83 - 1132 | 11.56 - 1492 | 2.35 - 943 |
> | Phi-3.5-mini | Brief | 70.89 - 282 | 56.76 - 420 | 33.09 - 1143 | 10.24 - 1805 | 3.00 - 984 |
> | Gemma3-1B | Normal | 42.15 - 325 | 19.58 - 427 | 36.40 - 1042 | 12.40 - 1414 | 2.25 - 2279 |
> | Gemma3-1B | Brief | 41.47 - 319 | 19.38 - 427 | 35.06 - 1078 | 12.47 - 1445 | 2.35 - 2198 |
>
> 2. **SAR does not over-penalize necessary derivations, since:**
>
> - **Correctness reward is prioritized.** In our combined objective, correctness reward carries higher weight than SAR. If a longer derivation is necessary to reach the correct answer, the correctness component will dominate, and such derivations will receive a high total reward.
> - **SAR is content-sensitive.** When intermediate reasoning is tightly related to the query (i.e., $P(a \mid q) \gg P(a)$), SAR assigns a *high* score even for longer outputs (see **Sec. 4.2**). Empirically, **Table 2** shows that Type 3 answers (longer but sensical incorrect attempts) are ranked above Type 4 under SAR, so SAR values quality of reasoning, not length alone.
>
> 3. **Token reduction primarily removes non-informative content.**
>
> Our analysis (see **Table 6**) shows only slight reductions in core reasoning behaviors for SAR-trained models compared to RL baselines. The majority of token savings come from removing **non-informative content** (e.g., verbose transitions, filler phrases), not from truncating essential reasoning steps. In **App. F.2** we provide representative examples contrasting GRPO and SA-GRPO outputs to illustrate where tokens are saved.

---

> ### Author Response · Authors · 2025-11-22
>
> **Q6: Can token-level SAR scores be calculated to make the metric more robust?**
>
> While the token-level \(v\)-score introduced in **Sec. 4.2** helps explain the mechanism of SAR, we intentionally use **sequence-level** rewards during RL training. There are two main reasons:
>
> - Some prevalent RL frameworks for LLMs—namely GRPO—are mainly built around sequence-level rewards, making SAR directly applicable without redesigning the training loop. This preserves comparability with existing methods and avoids confounding effects from token-level credit assignment.
> - Token-level \(v\)-scores alone can be noisy and do not reliably capture the *importance* of each token. For example, in **Figure 2**, the final answer token (“18”) has a low \(v\)-score even though it is crucial. Aggregating \(v\)-scores across the full sequence smooths out these local variances and yields a more stable and semantically aligned training signal.
>
> ---
>
> **Q7: What happens if the query contains spurious details?**
>
> We appreciate the reviewer raising this subtle point. As noted in **Lines 242–246**, SAR assigns high reward when the model “drawing on information from the question for the first time tend to receive high scores”. Because SAR does not explicitly distinguish “useful” from “irrelevant” query details, it may reward copying spurious content. For example, adding an unrelated phrase like *“Today is a sunny day”* could increase SAR if the model repeats it.
>
> We acknowledge this limitation. Nevertheless, the benchmarks we use in the paper are free from such problem. All datasets used are carefully constructed reasoning tasks in which every piece of provided information is intended to be relevant. In such settings, SAR’s assumption aligns with standard evaluation practice. We leave detecting irrelevant content in the query to future work.
>
> ---
>
> **Q8: Do results hold when the base model is much stronger?**
>
> Due to constrains of computation, the largest model we could afford in the experiment is 4B. However, we believe self-aligned reward will still lead to substantial performance gains for larger models, since **gains from RL scale with base-model capability.** RL improves performance by selecting trajectories that lead to more desirable outcomes, and the *quality* of these trajectories depends heavily on the base model. Prior work **[2][3]** shows that RL primarily amplifies the underlying capabilities of the model. Consequently, a stronger base model actually has *greater room* to benefit from RL-based optimization.
>
> ---
>
> We value your insightful comments and suggestions, which have significantly contributed to the improvement of our paper. We hope that our detailed responses and additional experiments have effectively addressed your concerns. If you find that your concerns have been resolved, we would be very grateful if you could reconsider the review score.
>
> **References**
>
> **[1]** The Overthinker's DIET: Cutting Token Calories with DIfficulty-AwarE Training
>
> **[2]** Cognitive Behaviors that Enable Self-Improving Reasoners, or, Four Habits of Highly Effective STaRs
>
> **[3]** Does Reinforcement Learning Really Incentivize Reasoning Capacity in LLMs Beyond the Base Model?, NeurIPS25

---

> > ### Comment · Reviewer_F8KC · 2025-11-27
> >
> > Thanks for the responses. I have already raised the score.

---

### Official Review · Reviewer_vsNW · 2025-10-28

**Soundness:** 3
**Presentation:** 4
**Contribution:** 3
**Rating:** 8
**Confidence:** 4

**Summary:**

The paper introduces "self-aligned reward", the difference in perplexity on the generated solution with and without the query in-context, as an additional reward term to binary verifiable rewards in order to incentivise concise, query-specific reasoning. The authors motivate SAR through a case study that connects the GRPO advantages assigned to responses with various response types and that analyses the token-level contributions to the overall SAR term. The paper demonstrates that using SAR for training with PPO or GRPO leads to a Pareto improvement in token efficiency and accuracy. An ablation study on the reward terms, including a comparison to entropy regularisation, demonstrates the importance of the complete proposed reward combination. Additional analysis on qualitative reasoning behaviours and training costs are used to further illustrate the impact of SAR.

**Strengths:**

The paper effectively uses a case study to motivate the design decisions behind SAR, which provides an intuition for the subsequent results. The method itself convincingly improves on the length minimisation baselines in striking a balance between accuracy and token efficiency. The analysis section features appropriate ablations and the reasoning behaviour frequencies provide further evidence that reasoning traces from models without SAR or length penalties are bloated with subsequences that do not correspond core reasoning behaviours. The paper is well-written and results are presented clearly.

**Weaknesses:**

Using GPT-4o to annotate responses without checking the quality of these annotations, such as by comparing to gold-standard human annotations on a subset, means that the corresponding analysis sections cannot be taken to be fully reliable.

It would be useful to show AES for a few different values of gamma. This would compliment the following section on accuracy-efficiency tradeoff nicely.

Whilst it's intuitive that entropy regularisation in the reward can lead to overconfidence and entropy collapse, explicitly verifying that using delta perplexity (SAR) mitigates this would support this claim, which is currently based only on accuracy differences.

**Questions:**

Do you have an intuition for why SA-GRPO leads to considerably more backtracking behaviour on NuminaMATH and AIME?

---

> ### Author Response · Authors · 2025-11-22
>
> We're very grateful for your constructive comments, and are glad for your acknowledgement of our paper’s clear motivation and thorough experiments. Below are responses to your suggestions and concerns.
>
> ---
>
> **Q1: In Section 4, how can the GPT-4o annotations be verified?**
>
> We thank the reviewer for raising this concern. Although the annotation task is relatively simple, we agree that evaluating on a human-annotated test set would further strengthen the reliability of the experiment. We will consider adding such an evaluation in the revised version.
>
> Our statement that “the task is rather simple” refers to the limited distinctions that GPT-4o needs to make:
>
> - **Types 5 and 6** are artificially created and come with fixed labels.
> - For *correct* answers, GPT-4o only needs to distinguish between **Type 1 vs. Type 2**, which amounts to detecting whether redundant content appears in the response.
> - For *incorrect* answers, GPT-4o only needs to distinguish **Type 3 vs. Type 4**, which amounts to checking whether *any* part of the reasoning is sensible.
>
> Given these straightforward criteria, we believe the annotation task is well within GPT-4o’s capabilities, though we agree that complementary human evaluation would improve robustness.
>
> ---
>
> **Q2: It would be useful to show AES for a few different values of gamma.**
>
> We show the AES values for different gamma values, based on the results in **Table 3**. Generally speaking, SAR has more significant advantages when gamma is higher (the scenario focuses more on accuracy). In addition, as a method that improves accuracy and efficiency simultaneously, the AES value of SAR exceeds baselines under several gamma values.
>
> | Model | Method | **1** | **3** | **5** | **8** |
> | --- | --- | --- | --- | --- | --- |
> | Qwen3-1.7B | GRPO-O1 | **0.580** | 1.11 | 1.65 | 2.46 |
> | Qwen3-1.7B | GRPO-ER | 0.459 | 0.96 | 1.47 | 2.24 |
> | Qwen3-1.7B | SA-GRPO | 0.577 | **1.19** | 1.79 | 2.71 |
> | Qwen3-4B | GRPO-O1 | 0.483 | 1.03 | **1.59** | 2.43 |
> | Qwen3-4B | GRPO-ER | **0.514** | **1.05** | **1.59** | 2.40 |
> | Qwen3-4B | SA-GRPO | 0.178 | 0.87 | 1.56 | **2.60** |
> | Phi-3.5-mini | GRPO-O1 | 0.694 | 1.00 | 1.31 | 1.78 |
> | Phi-3.5-mini | GRPO-ER | 0.717 | 1.04 | 1.36 | 1.85 |
> | Phi-3.5-mini | SA-GRPO | **0.881** | **1.50** | **2.13** | **3.08** |
> | Gemma3-1B | GRPO-O1 | 0.000 | 0.759 | 1.53 | 2.68 |
> | Gemma3-1B | GRPO-ER | 0.260 | 1.03 | 1.80 | 2.96 |
> | Gemma3-1B | SA-GRPO | **0.502** | **1.36** | **2.21** | **3.50** |
>
> ---
>
> **Q3: why does the author claim SAR mitigate the overconfidence and entropy collapse questions, besides the increased accuracy?**
>
> Prior work **[3][4]** has shown that entropy-minimization–based objectives can lead to **entropy collapse**, which causes the policy to converge prematurely and prevents the model’s full potential from being realized **[5]**.
>
> We claim that SAR mitigates this issue for two main reasons:
>
> 1. **SAR does not directly minimize perplexity.** Instead of rewarding low perplexity, SAR maximizes the *difference* between the perplexity with the question and the perplexity without the question, which preserves flexibility in the policy rather than forcing deterministic behavior.
> 2. **Empirically, the entropy trajectory of SA-GRPO matches that of rule-based GRPO.** During training, the entropy curve of SA-GRPO behaves similarly to that of the ground-truth–supervised GRPO baseline, rather than decreasing sharply. This indicates that SAR does not accelerate entropy decay and therefore does not induce entropy collapse.
>
> Together, these observations suggest that SAR improves accuracy without triggering the negative optimization dynamics typically associated with entropy-minimization methods.
>
> ---
>
> **Q4: Why SA-GRPO leads to considerably more backtracking behaviour on NuminaMATH and AIME?**
>
> Recent studies **[1][2]** indicate behaviors like backtracking are more common on challenging benchmarks. NuminaMATH and AIME are harder than other benchmarks, so the model may nee to backtrack more often to solve the problems.
>
> ---
>
> We value your insightful comments and suggestions, which have significantly contributed to the improvement of our paper. We hope that our detailed responses and additional experiments have effectively addressed your concerns.
>
> **References**
>
> **[1]** Cognitive Behaviors that Enable Self-Improving Reasoners, or, Four Habits of Highly Effective STaRs
>
> **[2]** Simplerl-zoo: Investigating and taming zero reinforcement learning for open base models in the wild
>
> **[3]** No Free Lunch: Rethinking Internal Feedback for LLM Reasoning
>
> **[4]** The Unreasonable Effectiveness of Entropy Minimization in LLM Reasoning
>
> **[5]** The Entropy Mechanism of Reinforcement Learning for Reasoning Language Models

---

### Official Review · Reviewer_SNsy · 2025-10-30

**Soundness:** 3
**Presentation:** 3
**Contribution:** 2
**Rating:** 6
**Confidence:** 3

**Summary:**

To address the overthinking or repetitive reasoning in large language models, this work proposes a self-aligned reward combined with a verifiable reward to enhance the accuracy and efficiency of the mathematical reasoning capabilities of LLMs. The self-aligned reward $R_{SA}$ is based on the perplexity drop from the answer's stand-alone perplexity to the perplexity when the answer is conditioned on the query $ppl(a) - ppl(a|q)$. In other words, if the answer is highly dependent on the query (not generic tokens), then the difference will be high, and the model can receive a larger self-aligned reward.

The proposed reward is applied with PPO and GRPO for 4 different LLMs across 5 mathematical reasoning datasets and 2 logical reasoning datasets. Detailed ablation studies are also conducted, indicating the importance of applying both the verifiable and self-aligned reward at the same time, and the effectiveness of the perplexity-based reward can reduce the output sequence length.

**Strengths:**

* The proposed reward function is well motivated. Measuring the difference between the stand-alone perplexity of an answer and its perplexity conditioned on the query provides a reasonable approach to capturing trivial/generic reasoning steps. Combining the verifiable (external) reward with the intrinsic reward signal effectively optimises the model for both accuracy and efficiency.
* The proposed reward signal is integrated with PPO and GRPO to optimise four different LLMs (two Qwen variants, Phi-3.5, and Gemma3), and is evaluated across seven diverse datasets covering mathematical and logical reasoning tasks.

**Weaknesses:**

* The idea of combining a verifiable reward with a length penalty is well aligned with recent developments in reinforcement learning for reasoning optimisation. However, this topic has been widely studied in prior work as well. For example, T1 [1] integrates correctness with a rule-based negative reward for undesirable reasoning behaviours, ThinkPrune [2] introduces an iterative pruning strategy to encourage concise reasoning, and Self-Braking Tuning [3] employs a frequency-based masking approach to mitigate overthinking during supervised training. A more detailed comparison with these approaches, particularly T1, would substantially strengthen the paper’s contribution and clarify how the proposed method advances beyond existing studies.
* A statistical significance test is missing from the experimental results. Some reported improvements appear marginal, for instance, in Table 3, the average accuracy of GRPO and SA-GRPO is 52.49 and 54.13, respectively. Without a significance test, it is difficult to determine whether these improvements are statistically meaningful. Including such tests would enhance the reliability of the evaluation.
* The Entropy method [4], which is listed as one of the main related works in Table 1, is only included in the ablation study (Section 6.1 and Table 5) but not in the main comparative results (Sections 5 and 6.2). Given its conceptual relevance, excluding Entropy from the main comparison raises concerns about fairness, as the proposed method is otherwise compared primarily against weaker length-penalty baselines. A more comprehensive comparison would provide a clearer picture of the model’s relative performance.
* The authors claim that the intrinsic reward signal is less prone to reward hacking compared to external signals (L067–068). However, this claim is not sufficiently supported by experimental evidence. In fact, the results suggest the opposite: the accuracy of the model optimised with the intrinsic reward ($R_{SA}$) drops from 69.07 to 20.96 on average (Table 5), indicating that the model may be exploiting or “hacking this reward signal. An additional experiment or analysis demonstrating the robustness of the intrinsic reward would be necessary to substantiate the authors’ argument.

[1] T1: Advancing Language Model Reasoning through Reinforcement Learning and Inference Scaling (Hou et al., ICML 2025)

[2] ThinkPrune: Pruning Long Chain-of-Thought of LLMs via Reinforcement Learning (Hou et al., arXiv 2025)

[3] Self-Braking Tuning: Let LRMs Break Free from Overthinking via Self-Braking Tuning (Zhao et al., NeurIPS 2025)

[4] Entropy: The Unreasonable Effectiveness of Entropy Minimization in LLM Reasoning (Agarwal et al., arXiv 2025)

**Questions:**

* Section 6.3 (Training cost of self-aligned reward) could move to the appendix. On the other hand, some interesting contents in the Appendix, e.g. the analysis of the self-aligned reward function in Appendix D.2.1 or experiments of vLLMs in Appendix E.2, could move to the main text.

---

> ### Author Response · Authors · 2025-11-22
>
> We're very grateful for your constructive comments, and are glad for your acknowledgement of our paper’s clear motivation and thorough experiments. Below are responses to your suggestions and concerns.
>
> ---
>
> **Q1:  A more detailed comparison with relevant approaches would substantially strengthen the paper’s contribution and clarify how the proposed method advances beyond existing studies.**
>
> We thank the reviewer for highlighting these relevant works. We will incorporate a more detailed comparison in the revised **Related Work** section. At a high level, the novelty of our method lies in proposing a **content-aware reward metric** that explicitly encourages concise yet complete reasoning, thereby improving **both accuracy and efficiency** (see **Table 1** for a summary of our method’s properties). Regarding the works mentioned by the reviewer:
>
> - **T1** adopts multiple advanced techniques to achieve strong performance, but it does not specifically target efficiency. Its overlength penalty mainly prevents infinite looping and is not a strong or content-sensitive efficiency signal.
> - **ThinkPrune** is conceptually aligned with O1 and ER (discussed in our paper): it gradually encourages shorter chains of thought through length penalization. However, the penalty is **content-agnostic**, which may prune essential reasoning steps and thus risks harming correctness.
> - **SBT** lies in a different research direction, focusing on SFT rather than RL. Our method is orthogonal to SBT and can likely be combined with it to further improve efficiency. In addition, both ThinkPrune and SBT show up to ~5% performance degradation on some models.
>
> ---
>
> **Q2: Are the experimental results statistically significant?**
>
> We thank the reviewer for raising this important point. We perform a one-sided McNemar test to compare **PPO vs. SA-PPO** and **GRPO vs. SA-GRPO** in **Table 3**. The results are provided below. Here, **1** indicates that the self-aligned reward (SAR) is statistically better (p < 0.05), **–1** means the baseline is statistically better, and **0** means no significant difference.
>
> Across all settings, SAR shows statistically significant improvements in **45%** of test cases and maintains *at least comparable* performance in **95%** of cases. Given that SAR reduces answer length by **~30% on average**, these results clearly show that SAR achieves a **Pareto improvement** in both accuracy and efficiency.
>
> | Model | Algorithm | GSM8k | GSM-symbolic | MATH | NuminaMath | AIME |
> | --- | --- | --- | --- | --- | --- | --- |
> | Qwen3-1B | PPO | 1 | 1 | 1 | 1 | 0 |
> | Qwen3-1B | GRPO | 0 | 0 | 1 | 1 | 0 |
> | Qwen3-4B | PPO | 1 | 0 | -1 | 0 | 1 |
> | Qwen3-4B | GRPO | 0 | 1 | 0 | 1 | 1 |
> | Phi3.5-mini | PPO | 1 | 1 | 1 | 1 | 0 |
> | Phi3.5-mini | GRPO | 0 | 0 | 1 | 0 | 0 |
> | Gemma3-1B | PPO | 0 | 1 | 0 | -1 | 0 |
> | Gemma3-1B | GRPO | 1 | 0 | 1 | 0 | 0 |
>
> ---
>
> **Q3: Why is the Entropy method only included in the ablation study part (but not the main table)?**
>
> The entropy-based baseline **[1]** is indeed closely related. We include it in the ablation study for two main reasons:
>
> 1. The original entropy method is fully unsupervised—it does not use ground-truth labels. Recent work **[2]** has questioned the effectiveness of such unsupervised reward designs, so this is not yet a fully accepted method. In our **Table 5 (rows 3–4)**, we reproduce the original entropy approach and find that it offers only limited performance gains.
> 2. We structured the experiments so that each section highlights a distinct message. Since the main contribution of our paper is a reward that jointly improves accuracy and efficiency, the main experiment in **Sec. 5** focuses on comparing SAR with efficiency-oriented RL algorithms. In contrast, **Sec. 6.1** centers on reward design. The ablation study there clarifies:
>     - why incorporating ground truth is still valuable;
>     - Why using the *difference between two perplexities* (**Lines 430–431**) is a principled choice.
>
>     Because entropy-based methods are not designed for efficiency, we include them in Sec. 6.1, which is dedicated to reward formulation rather than efficiency comparison.

---

> ### Author Response · Authors · 2025-11-22
>
> **Q4: Why does the authors claim the intrinsic reward signal is less prone to reward hacking compared to external signals?**
>
> We appreciate the reviewer’s question and would like to clarify a potential ambiguity in our original writing. In our paper, **“external reward” specifically refers to reward modeling approaches**, i.e., training an LLM to act as a learned reward evaluator. It would be more precise to simply refer to this category as *reward modeling*, since **rule-based ground-truth signals are not considered “external rewards”** in our terminology.
>
> While reward models are well-known to be susceptible to reward hacking, we fully acknowledge that **rule-based ground truth remains the most reliable source of supervision**. Our self-aligned reward (SAR) is *not* designed to replace these signals; rather, **SAR complements ground-truth rewards by encouraging reasoning that is both more efficient and more faithful to the model’s own internal consistency**. We will refine the terminology in the revised version to avoid confusion and to more clearly distinguish between reward modeling and ground-truth supervision.
>
> ---
>
> **Q5: The arrangement of main sections / appendix sections?**
>
> We thank the reviewer for the helpful suggestions. We do have some interesting experiments in the appendix sections, but they aren’t moved into the main paper due to space constraints. In the revised version (which has 1 extra page), we will re-arrange the sections to make sure the main paper contains all the crucial information.
>
> ---
>
> We value your insightful comments and suggestions, which have significantly contributed to the improvement of our paper. We hope that our detailed responses and additional experiments have effectively addressed your concerns. If you find that your concerns have been resolved, we would be very grateful if you could reconsider the review score.
>
> **References**
>
> **[1]** The Unreasonable Effectiveness of Entropy Minimization in LLM Reasoning
>
> **[2]** No Free Lunch: Rethinking Internal Feedback for LLM Reasoning

---

> > ### Comment · Reviewer_SNsy · 2025-11-26
> >
> > Thank you for your response. As I have already given a high score, I would like to keep it unchanged.

---

### Official Review · Reviewer_oz4c · 2025-11-01

**Soundness:** 3
**Presentation:** 3
**Contribution:** 3
**Rating:** 8
**Confidence:** 4

**Summary:**

The paper aims to improve the conciseness of LLM outputs without sacrificing correctness. It proposes self-aligned reward (SAR), which considers the difference between the perplexity of answers when conditioned vs. not conditioned on the query. In contrast, existing approaches penalize the length of answers without concern for relevance to the query. SAR provides a Pareto improvement over baselines, improving both accuracy and conciseness.

**Strengths:**

The paper has a clear motivation, improving conciseness of LLM answers, and provides a simple intuitive solution, expressed as a reward function in (6).

Experiments are very comprehensive, considering a variety of models, datasets, and metrics. I especially appreciated the plots in Figure 3, showing performance gain vs. length reduction. It supports the main argument of the paper well: SAR results in the biggest performance gain across all length reduction levels.

Besides the gains in efficiency, the example in Section 4.2 illustrates that SAR could even be useful as an interpretability technique. It shows how SAR can be used to rank the relevancy of each output token to a given query.

**Weaknesses:**

In Section 4.1, evaluating different types of answers using different reward functions to see what SAR captures that alternative reward functions don’t is a great idea. The main difference seems to be that SAR is the only reward function that penalizes answers of type “Correct but no though”. However, how this type of answers are generated is not explained well in the text (“artificially synthesized to simulate memorization” is not a sufficient explanation). Without clarity on what this answer type stands for, the results in Section 4.1 are not helpful.

Columns in Table 1, “Continuous”, “Internal”, “Content-Aware”, “Correctness”, and “Conciseness”, are not defined. It is ambiguous what criteria is meant.

**Questions:**

How do (5) in Section 4.1 “simulate memorization”?

---

> ### Author Response · Authors · 2025-11-22
>
> We're very grateful for your constructive comments, and are glad for your acknowledgement of our paper’s clear motivation and strong results. Below are responses to your suggestions and concerns.
>
> ---
>
> **Q1: In Section 4.1, how are different types of answers generated? What’s the main difference between SAR and other rewards?**
>
> Answer types 1–4 are generated by sampling candidate responses from Qwen3-4B and then annotating them using GPT-4o. Types 5–6, on the other hand, are *manually crafted* to simulate **memorization-style responses**. This design reflects the behavior of an LLM that has memorized the answer: instead of showing any reasoning process, it simply outputs the final answer. An example is provided in **Table 12:**
>
> ```
> For a question with an answer of 18:
> Type 5 (Correct, Memorized): The answer is 18.
> Type 6 (Incorrect, Memorized): The answer is 0.
> ```
>
> As shown in **Table 2**, SAR exhibits three desirable properties that make it a strong indicator of answer quality. In particular, beyond **penalizing Types 5 and 6** (memorization-like outputs), SAR also successfully enforces the intended ranking structure:
>
> - **Type 1 > Type 2**
> - **Type 3 > Type 4**
>
> These ranking patterns reflect SAR’s ability to distinguish nuanced differences in reasoning quality, not merely correctness, making it more informative than standard rewards.
>
> ---
>
> **Q2: Please clearly define the columns in Table 1.**
>
> In **Table 1**, different columns refer to favorable features of RL rewards:
>
> - Continuous: The reward has a continuous range of values, instead of discrete values.
> - Internal: The reward can be calculated without a reward model.
> - Content-Aware: The reward considers the quality of the generated content.
> - Correctness / Conciseness: The reward favors correct / short answers.
>
> Thanks for pointing out this detail, and we’ll surely add the clarifications in the revised version.
>
> ---
>
> We value your insightful comments and suggestions, which have significantly contributed to the improvement of our paper. We hope that our detailed responses and additional experiments have effectively addressed your concerns.

---

### Meta-Review · Area_Chair_Gmqg · 2025-12-14

**Summary:**

The reviewers raised several concerns regarding the paper’s clarity and experimental rigor. Reviewer oz4c pointed out ambiguities in Section 4.1, specifically regarding the generation process of various answer types used for evaluation. Reviewer SNsy highlighted: 1) insufficient discussion and comparison with related works, 2) lack of statistical significance analysis, 3) absence of an entropy minimization baseline, and 4) missing analysis on the robustness of the self-alignment reward. Reviewer vsNW questioned the quality of GPT-4o-generated annotations, requested an ablation study on the hyperparameter $\gamma$, and sought evidence that "delta perplexity" effectively mitigates overconfidence and entropy collapse. Reviewer F8KC noted the lack of adversarial experiments regarding RSA's role in reasoning trace quality, potential RL training instability (due to KL divergence fluctuations), shifting reward scales, and inconsistencies in experimental setups compared to prior work.

**Reviewer Concerns:**

The concerns from Reviewer oz4c were relatively minor and primarily centered on the clarity of Section 4.1. the AC believes the authors' revisions have successfully addressed these points. Reviewers SNsy and F8KC actively participated in the discussion and maintained their positive scores. For Reviewer vsNW, the authors provided a comprehensive response and new quantitative results. The AC found these additional results compelling and considers the previous concerns well-addressed.

**Reviewer Scores:**

Reviewer oz4c is expected to maintain a positive score as the clarity issues were resolved. Reviewers SNsy and F8KC confirmed their positive assessments during the discussion. Given the authors provided the requested experimental data, the AC believes Reviewer vsNW would also maintain or confirm a positive rating.

---

### Decision · Program_Chairs · 2026-01-26

Accept (Poster)